# Predicting the Performance of Foundation Models via Agreement-on-the-Line

**Rahul Saxena**[*1]   **Taeyoun Kim**[*1]   **Aman Mehra**[*1]   **Christina Baek**[1]
**Zico Kolter**[1,2]   **Aditi Raghunathan**[1]

Carnegie Mellon University[1], Bosch Center for AI[2]
`{rsaxena2, taeyoun3, amanmehr, kbaek, zkolter, raditi}@cs.cmu.edu`

## Abstract

Estimating the out-of-distribution performance in regimes where labels are scarce is critical to safely deploy foundation models. Recently, it was shown that ensembles of neural networks observe the phenomena "agreement-on-the-line", which can be leveraged to reliably predict OOD performance without labels. However, in contrast to classical neural networks that are trained on in-distribution data from scratch for numerous epochs, foundation models undergo minimal finetuning from heavily pretrained weights, which may reduce the ensemble diversity needed to observe agreement-on-the-line. In our work, we demonstrate that when lightly finetuning multiple runs from a *single* foundation model, the choice of randomness during training (linear head initialization, data ordering, and data subsetting) can lead to drastically different levels of agreement-on-the-line in the resulting ensemble. Surprisingly, only random head initialization is able to reliably induce agreement-on-the-line in finetuned foundation models across vision and language benchmarks. Second, we demonstrate that ensembles of *multiple* foundation models pretrained on different datasets but finetuned on the same task can also show agreement-on-the-line. In total, by careful construction of a diverse ensemble, we can utilize agreement-on-the-line-based methods to predict the OOD performance of foundation models with high precision.

## 1   Introduction

Foundation models (FM), or large models first pretrained on open world data then finetuned or prompted for a specific downstream task, have proven to be powerful solutions for many common machine learning problems. A notable trait about FMs is that they are far more robust to distribution shift than other deep learning approaches — across image and language benchmarks, they suffer a smaller performance degradation on out-of-distribution (OOD) data, that may vary substantially from the in-distribution (ID) finetuning data [43, 42, 7, 60, 58, 14]. From clinical decision-making in different hospitals to navigating robots through unseen terrains, FMs are increasingly utilized for tasks prone to distribution shift. However, evaluating these models in OOD settings remains difficult: in many cases, acquiring labels for OOD data is costly and inefficient, while unlabled OOD data is much easier to collect. Although the field has explored other means for estimating OOD accuracy without labeled data, they are not ideal for FMs. A reliable FM performance estimator has the following desirable properties. First, the method must be computationally efficient to account for FMs' large model size. Second, FMs are leveraged for many different tasks (e.g., classification, question-answering, regression), so the method should also be versatile across tasks. Third, as we will see, methods for finetuned FMs may require *different model assumptions from neural networks trained from scratch*.

38th Conference on Neural Information Processing Systems (NeurIPS 2024).

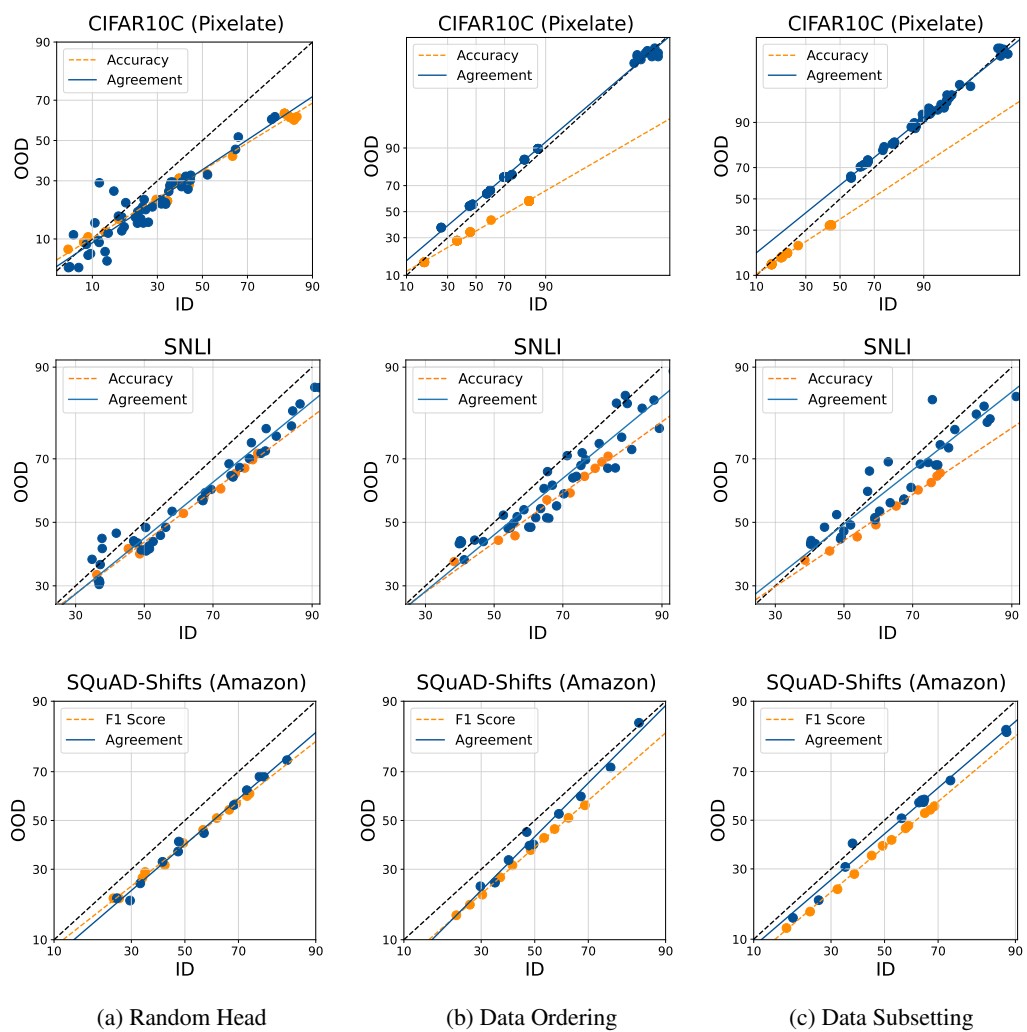

Figure 1: The ID vs OOD lines for accuracy (orange) and agreement (blue) for various datasets and fine-tuned ensembles. Each blue dot corresponds to a member of the ensemble and represents the ID (x) and OOD (y) accuracy. Each orange dot corresponds to a pair of these members and represents the ID (x) and OOD (y) agreement. From CIFAR10 to CIFAR10C "Pixelate" in linear probed CLIP, MNLI to SNLI in full fine-tuned OPT, and SQuAD to SQuAD-Shifts "Amazon" in full fine-tune GPT2, we observe that randomly initializing the head as the diversity source for generating ensembles (columns) shows the closest agreement linear fit to accuracy.

Recently, [2] proposed a promising method for estimating the OOD accuracy of deep networks using the *agreement* between pairs of these classifiers (i.e., how often two classifiers make the same prediction). For distribution shifts where models observe a strong linear correlation in ID versus OOD accuracy – a common phenomenon in vision and language benchmarks [39, 1] – a strong linear correlation also holds for ID versus OOD agreement with extremely similar slopes and intercepts. These effects are referred to as accuracy-on-the-line (ACL) and agreement-on-the-line (AGL) respectively, and together they provide a simple method for estimating OOD accuracy via unlabeled data alone. Namely, without any OOD labels, we can instead measure the linear fit of ID versus OOD agreement as a proxy for the linear fit of ID versus OOD accuracy. With this linear fit, we can verify whether ACL holds by using the correlation strength of agreement's linear trend and estimating each model's OOD accuracy by linearly transforming ID accuracy. This simple approach has shown to reliably predict the OOD accuracy of models within a few percentage points across classification and question-answering tasks.

Unfortunately, while the method has several practical advantages, it is unclear whether finetuned FMs also observe the necessary AGL phenomena. Intuitively, a prerequisite to observing AGL is a *diverse ensemble* of classifiers. Since OOD accuracy falls below ID accuracy, if the linear trend in ID versus OOD agreement is to match that of accuracy, models must also *agree much less OOD than ID*. For this to happen, errors between any two models must be sufficiently decorrelated. [2] observes AGL in ensembles of neural networks trained for hundreds of epochs from scratch where it is conceivable that the stochasticity between training runs leads to large divergences in the weight space, and corresponding models have diverse OOD predictions. However, in the case of finetuned FMs, models are much closer in the weight space. FMs are often either linear probed over the same pretrained weights or full finetuned for a few epochs with a small learning rate and intuitively, such *light* finetuning may lead to models that "revert back" to their pretrained behavior to make highly correlated predictions OOD.

This raises the question: can we enforce AGL in this paradigm of lightly finetuning heavily pretrained models? In this work, we conduct an extensive study across several modalities, e.g., CLIP-based image classification and LLM-based question-answering, and training regiments, e.g., full finetuning and linear probing, to understand when AGL holds for finetuned FMs. We first investigate whether AGL appears in an ensemble of finetuned models from a *single* base FM. To collect a deep ensemble, the following sources of diversity can be injected into the finetuning process: 1) random initialization of the linear head; 2) random data ordering; and 3) random data subsetting. We find that not every source of diversity during fine-tuning on ID data manifests in sufficient diversity OOD, breaking the matching linear fits in ID versus OOD accuracy and agreement. Interestingly, finetuning models from *different random initializations of the linear head consistently induces AGL* across benchmarks. In contrast, neural networks trained from scratch observe AGL irrespective to these diversity sources.

Second, we show that finetuned models from *multiple* different base FMs can be leveraged for AGL-based performance estimation. As base FMs can be pretrained with different datasets, architectures, and training regiments, the linear trends in ID versus OOD accuracy and agreement may break altogether in such ensembles. Indeed previous works indicate that on vision tasks, FMs pretrained on different image corpora can have different levels of OOD robustness for the same ID performance [17, 43, 51]. On the contrary, we find that on language tasks, FMs pretrained on different text corpora observe both AGL and ACL across question-answering and text classification tasks.

In total, we develop simple techniques for applying AGL-based performance estimation methods to predict the OOD performance of foundation models. We demonstrate that the AGL phenomenon is not limited to ensembles of neural networks trained from scratch. By simply finetuning FMs from random initializations of the linear head, we can observe the phenomena in FMs across a wide variety of tasks (classification, question-answering) and modalities (vision, language) and training procedures (linear probing, full finetuning). We find that AGL is the only method to accurately estimate the performance of finetuned FMs across all tasks, surpassing other performance estimation baselines by a significant margin as large as $20\%$ mean absolute percentage error.

## 2 Background and related work

### 2.1 Setup

We are interested in evaluating models that map an input $x \in \mathbb{X}$ to a discrete output $y \in \mathbb{Y}$. In particular, we finetune foundation models. For a base model B, let $f(\mathsf{B})$ denote a finetuned version of B. In this work, we consider a variety of foundation models: GPT2 [42], OPT [65], Llama2 [53], BERT [14], and CLIP [43].

**Finetuning strategies.** We have access to labeled data from some distribution $\mathcal{D}_{\mathrm{ID}}$ that we use for obtaining $f(\mathsf{B})$ from B. In this work, we consider the following standard finetuning procedures.

1. **Linear probing (LP):** Given features from the base model $\mathsf{B}_\theta$, we train a linear head $v$ such that the final classifier maps the score $v^\top \mathsf{B}_\theta(x)$ to a predicted class. We randomly initialize $v$ and update $v$ via gradient steps on a suitable loss function. The base model parameters remain frozen. We refer to $v$ as either a linear probe (classification), or span prediction head (question-answering) depending on the task.

2. **Full finetuning (FFT):** We update all parameters of the backbone $B_\theta$ and the linear head $v$ *using a small learning rate*. When infeasible to update all parameters, we perform *low-rank adaptation* (LoRA) [25] to reduce the number of trainable parameters while still effectively updating the feature extractor $B_\theta$. In this work, we do not distinguish between LoRA and FFT as they conceptually achieve the same effect, and seem to show similar empirical trends in our studies.

**OOD performance estimation.** Given access to a labeled validation set from $\mathcal{D}_{\text{ID}}$ and *unlabeled* samples from a related but different distribution $\mathcal{D}_{\text{OOD}}$, our goal is to estimate performance on $\mathcal{D}_{\text{OOD}}$. We consider the standard performance metrics for various tasks: Accuracy $\ell_{0\text{-}1} : \mathbb{Y} \mapsto [0, 1]$ for classification, and Exact Match $\ell_{\text{EM}} : \mathbb{Y} \mapsto [0, 1]$ and Macro-averaged F1 score $\ell_{\text{F1}} : \mathbb{Y} \mapsto [0, 1]$ for question-answering. We use $\ell$ to denote the appropriate metric in the context.

## 2.2 Background on OOD accuracy estimation

There is rich literature on OOD performance estimation for deep networks, with a variety of proposed approaches. Initial works focused on upper bounding the degree of distribution shift through data and/or model dependent metrics, e.g., uniform convergence bounds using $\mathcal{H}$-divergence [4, 37, 11, 30]. However, these bounds tend to be loose for deep networks [39]. The following works try to estimate the performance exactly.

For classification, [23, 22, 19, 16, 21] leverage the model's confidence to predict the OOD performance. Since deep models are typically overconfident, these models are first calibrated in-distribution by temperature scaling. Similar methods are uncertainty quantification works that directly calibrate models under distribution shift [62, 67, 41]. Confidence based methods are commonly utilized in practice, and favorable for foundation models as they are computationally light and model-agnostic. However, they often fail for large shifts [19] and are often well-defined for accuracy but not other common metrics like F1 score. These can be limiting factors for foundation models which are applied to a broad array of tasks. Still, as they are the most common estimation methods, we utilize them as the baselines in our work.

[49, 12, 13] also measure model behavior on known auxiliary tasks to understand model behavior under the distribution shift at hand. However, these approaches tend to be overfit to specific datasets or modalities. Similar to AGL, there are prediction methods that utilize information from ensembles. Oftentimes a separate "reference" ensemble is trained on some objective to predict the performance of a "target" model [10, 63, 8]. These methods have a higher computational cost than AGL. Although AGL also requires at least 3 models to compute agreement, these models only undergo generic finetuning. Thus, it is a better suited approach for evaluating foundation models, especially if off-the-shelf finetuned models are readily available, e.g., from Huggingface (see Section 4).

Overall, there is growing attention towards understanding the safety and reliability of foundation models. To understand the effective robustness of FMs under distribution shift, recent works focus on studying the "accuracy-on-the-line" phenomena [39] (details in next subsection) and designing benchmarks that expose different failure modes of large models [36, 54]. However, unsupervised OOD performance estimation is underexplored in this modern setting, in terms of new methods and the transferability of old methods to large pretrained models.

## 2.3 Accuracy and agreement on the line

We are interested in adapting the method "agreement-on-the-line" (AGL) [2] for OOD estimation as it obtains state-of-the-art performance estimation across several distribution shifts. AGL is based on an earlier observation called "accuracy-on-the-line" (ACL) — across common distribution shift benchmarks, there is a strong linear correlation between the ID and OOD performance of models [39, 45–47, 61, 51, 38]. ACL can also be observed in FMs for image classification, e.g., CIFAR10C [22], ImageNetV2 [46], FMoW-WILDS [28], and question-answering, e.g., SQuAD-Shifts [38]. However, ACL does not always hold, e.g., Camelyon-WILDS [39] and SearchQA [1].

While ACL is a striking phenomenon, it does not immediately provide a practical method to estimate OOD performance—computing the linear fit of ID versus OOD accuracy requires labeled samples from $\mathcal{D}_{\text{OOD}}$. Alternatively, we can estimate this linear trend exactly using only the agreement between neural networks [2]. Formally, given a pair of models $f_1$ and $f_2$ that map inputs to labels, accuracy

and agreement is defined as

$$\mathsf{Acc}(f_i) = \mathbb{E}_{x,y\sim\mathcal{D}}[\ell(f_i(x),y)], \ \ \mathsf{Agr}(f_1,f_2) = \mathbb{E}_{x,y\sim\mathcal{D}}[\ell(f_1(x),f_2(x))], \tag{1}$$

where $\ell$ is the appropriate performance metric of interest. While accuracy requires access to the ground truth labels $y$, agreement only requires access to unlabeled data and a pair of models. [2] observes that when ID versus OOD accuracy is strongly linearly correlated between neural networks, i.e., ACL, then the ID versus OOD agreement of pairs of these models also observe a strong linear correlation with the *same* linear slope and bias. Furthermore, when accuracies do not show a linear correlation, agreements also do not. This coupled phenomenon is dubbed "agreement-on-the-line" (AGL).

To use AGL for OOD performance estimation, one may obtain the slope and bias of the agreement line with unlabeled data, and then estimate the OOD performance by linearly transforming the ID validation performance. Specifically, with a collection of models $\mathcal{F} = \{f_1, f_2, ..., f_n\}$, AGL suggests that ID versus OOD accuracy observe a strong linear correlation if and only if ID versus OOD agreement observes a strong linear correlation and when they do, the slopes and biases match: $\forall f_i, f_j \in \mathcal{F}$ where $i \neq j$

$$\Phi^{-1}(\mathrm{Acc}_{\mathrm{OOD}}(f_i)) = a \cdot \Phi^{-1}(\mathrm{Acc}_{\mathrm{ID}}(f_i)) + b$$
$$\Updownarrow \tag{2}$$
$$\Phi^{-1}(\mathrm{Agr}_{\mathrm{OOD}}(f_i, f_j)) = a \cdot \Phi^{-1}(\mathrm{Agr}_{\mathrm{ID}}(f_i, f_j)) + b$$

$\Phi^{-1}$ is the probit transform used to induce a better linear fit as used in [2, 39]. Provided access to $\mathrm{Acc}_{\mathrm{ID}}(f_i), \mathrm{Agr}_{\mathrm{ID}}(f_i, f_j), \mathrm{Agr}_{\mathrm{OOD}}(f_i, f_j) \ \forall i, j$, we can estimate $\mathrm{Acc}_{\mathrm{OOD}}(f_i)$ for all $f_i \in \mathcal{F}$. We refer the reader to [2] for formal AGL-based performance estimation algorithms (ALine-S and ALine-D), which we also provide in Appendix A.1.1.

## 3 Predicting OOD performance: single base foundation model

We first evaluate whether AGL appears in an ensemble of multiple finetuned runs of a *single* base foundation model. This would enable precise OOD performance estimates for each ensemble member. A practitioner may naively gather a finetuned ensemble by training a couple runs with different seeds or hyperparameters. However, an overriding concern is that even with some randomness in the finetuning process, linear probing or light full-finetuning over the same base model may lead to solutions with very correlated predictions. We extensively evaluate the following methods of introducing diversity into the finetuning process to see what approach (if any) can lead to AGL.

1. **Random linear heads** We initialize the last layer of the network (i.e., the linear head) randomly, instead of via some zero-shot or pre-specified manner.

2. **Data ordering** We present the same training data to each model but shuffle the order of the data, i.e., model observes different minibatches.

3. **Data subsetting** We i.i.d. sample $p\%$ subset of the data to train over. In the main body, we report models trained on independently sampled $10\%$ of the training data, other proportions of $30\%$ and $50\%$ are reported in Appendix A.4.

We perturb one source of diversity at a time and study whether AGL occurs in each resulting model ensemble. For each setting, we also vary the number of training epochs to collect models with different ID performance, which is necessary to obtain a meaningful linear correlation in accuracy. The additional randomness induced by different training epochs does not affect the observations we make. We use at most four A6000's for all experiments except for linear probing where we use one RTX 8000.

### 3.1 VLM-based Image Classification

We first investigate the effect of diversity source on AGL behavior for vision benchmarks. For image classification, a common pipeline is to finetune over a CLIP [43] pretrained foundation model.

**CLIP Linear Probing**   We finetune over OpenCLIP ViT-B/32 model trained on LAION-2B [26]. Given its well-established zero-shot capabilities, a popular method of finetuning CLIP is to simply employ linear probing on top of the CLIP representation. We take particular interest in evaluating the OOD performance of an ensemble of linear models trained on top of frozen base model representations.

**Datasets**   We evaluate ensembles on synthetic corruptions (CIFAR10C, CIFAR100C, ImageNetC), dataset replication shifts (CIFAR10.1, ImageNetV2), style shifts (OfficeHome), geographical and temporal shifts (FMoW-WILDS, iWildCam-WILDS), and interlaboratory shifts in medicine (Camelyon17-WILDS). iWildCam-WILDS exhibits weak ACL and Camelyon17-WILDS doesn't exhibit any ACL [39]. We test on iWildCam-WILDS and Camelyon17-WILDS to verify AGL's negative condition, i.e., when the linear correlation does not exist in ID versus OOD accuracy, it also does not exist in agreement.

Table 1: We evaluate models on the following distribution shift benchmarks.

| ID | OOD |
|---|---|
| CIFAR10 [29] | CIFAR10C [22], CIFAR10.1 [45] |
| CIFAR100 [29] | CIFAR100C [22] |
| ImageNet [48] | ImageNetC [22], CIFAR10.1 [45] |
| FMoW ID [28] | FMoW OOD [28] |
| iWildCam ID [28] | iWildCam OOD [28] |
| Camelyon17 ID [28] | Camelyon17 OOD [28] |
| OfficeHome [56] | All (ID, OOD) pairings of domains Art, ClipArt, Product, Real World |
| MNLI [59] | MNLI-Mismatched [59], SNLI [6] |
| SQuAD [44] | SQuAD-Shifts [38] |

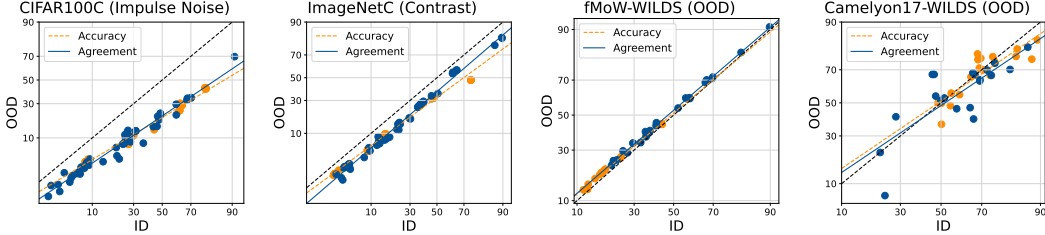

Figure 2: In ensembles with diverse random initializations, ACL and AGL holds across benchmarks in linear probed CLIP models. Similar to [2], neither ACL nor AGL holds for the Camelyon17-WILDS

**Results**   Across vision benchmarks, linear probed CLIP models observe ACL, i.e. there is a strong linear correlation in the ID versus OOD performance. Similarly, on the same datasets, we can observe a corresponding strong linear correlation in agreement across ensembles injected with diversity in linear head initialization, data ordering, and data subsetting (Figures 1). However, we find that only ensembles with diverse initialization leads to AGL where the linear trend of agreement and accuracy have a matching slope and bias (Figure 2). See Appendix A.3.2 for results on other datasets. In model ensembles obtained by data ordering and data subsetting, we observe a consistent trend where the agreement trend observe a much higher slope *close to the diagonal $y = x$ line*. These results are not specific to linear probing alone. In full finetuned CLIP models, we also observe that random linear heads induce the most reliable AGL behavior (See Appendix A.3). Note that this setting is still notably different from [2] where models are heavily trained for *tens to hundreds of epochs* often with a large learning rate, which causes AGL behavior to be more robust to the source of diversity used to induce the ensemble (See Appendix A.3.4).

## 3.2   LLM-based Question-Answering and Text Classification

We conduct a similar systematic investigation of AGL in finetuned runs of a single base *language* model. Similar to CLIP linear probing, we find that AGL cannot be observed without random head initialization in language models evaluated on text classification and extractive question-answering tasks. While we mostly focus on tasks that require a linear head during finetuning, we also conduct a diversity study on generative tasks where the base model is finetuned directly in Appendix A.6.1.

Table 2: ALine-D MAPE (%) of different sources of diversity for CLIP linear probing and GPT2-Medium/OPT-125M full finetuning. We average the score over all corruptions for CIFAR10C.

| Source of Diversity | CIFAR10C | SQuAD-Shifts Amazon | SQuAD-Shifts Reddit | SNLI |
|---|---|---|---|---|
| Random Linear Heads | **14.64** | **6.34** | **3.48** | **11.70** |
| Data Ordering | 37.01 | 10.30 | 9.59 | 15.40 |
| Data Subsetting | 35.85 | 16.21 | 13.94 | 15.50 |

**Full Finetuned Language Models**   We evaluate over a collection of 450 full finetuned runs of several base FMs: GPT2-Medium [42] and OPT-125M [65]. Models are full finetuned for up to 20 epochs with a small learning rate ($\leq 1e^{-6}$). Hyperparameters specifics can be found in Appendix A.2. We do not conduct a linear probing study for question-answering as it leads to poorly performing models. For text classification, we also conduct a linear probing study in Appendix A.5.

**Datasets**   We test models on a text classification shift from MNLI [59] in the GLUE benchmark [57] to MNLI-Mismatched [59] and SNLI [6]. We also evaluate extractive question-answering models on the shift from SQuAD v1.1 [44] to SQuAD-Shifts [38].

**Results**   We evaluate models on accuracy for text classification and F1 score for question-answering. Similar to our findings in CLIP, in both text classification and question-answering benchmarks, ensembles of full finetuned LLMs observe AGL when models are trained from different randomly initialized linear or span heads while data ordering and data subsetting observe an agreement trend closer to the diagonal $y = x$ line (Figure 1 and Appendix A.5). We note that with full finetuning, the differences in AGL behavior between diversity sources are not as stark as with linearly probed models. In some sense, how model diversity is achieved becomes increasingly less important for observing AGL as the base model parameters also diverge, with ensembles of models heavily trained from scratch at the extreme [2].

### 3.3   Summary and Implications

Across image and language modalities, we demonstrate that ensembles of finetuned FMs can also observe agreement-on-the-line similar to heavily trained CNN's [2]. In both domains and regardless of the fine-tuning strategy, e.g. FFT and LP, or metric, e.g. F1 and Accuracy, employed, the diversity induced via random head initialization yields AGL, while the diversity induced via data reordering or data subsetting does not. This phenomenon can be observed across hyperparameters (Appendix A.7) and different PEFT methods (Appendix A.8). With a single heavily pre-trained base FM, one may think that light finetuning leads to downstream models with highly correlated behavior under distribution shift. However, simply randomly initializing the linear head alone induces sufficiently decorrelated models for observing AGL. The diversity in the ensemble becomes important when predicting the OOD performance of models using downstream AGL-based methods. In Table 2, we show that AGL-based methods can only accurately predict the OOD performance of models in ensembles with diverse initialization, and cannot with data subsetting or ordering.

Furthermore, our findings contrast previous work that suggest AGL is a neural-network specific phenomenon [2, 31], unlike ACL which is model agnostic [39]. Specifically, [2] report that linear models trained on top of the flattened CIFAR10 images do not observe AGL. However, we find that, on top of CLIP features, *linear models can exhibit AGL* with random initialization. Previous work on the Generalization Disagreement Equality [27] contend that data subsetting leads to the most diversity in model predictions for deep ensembles (neural networks, random forests). Specifically, in-distribution, the agreement rate between pairs of models was shown to equal their expected accuracy in ensembles obtained by data subsetting, while those that vary random initialization has slightly higher agreement [27, 40]. On the other hand, in our problem setting of *out-of-distribution* datasets on FMs, we found that ensembles induced by different random initialization achieves AGL, while data ordering or subsetting cannot. Our setting is different from previous literature in two ways: (1) AGL studies the OOD agreement rate relative to their ID agreement, in contrast to the GDE phenomenon which only regards the models' ID agreement. We hypothesize that random initialization is much more important for observing the right levels of OOD agreement. (2) Models

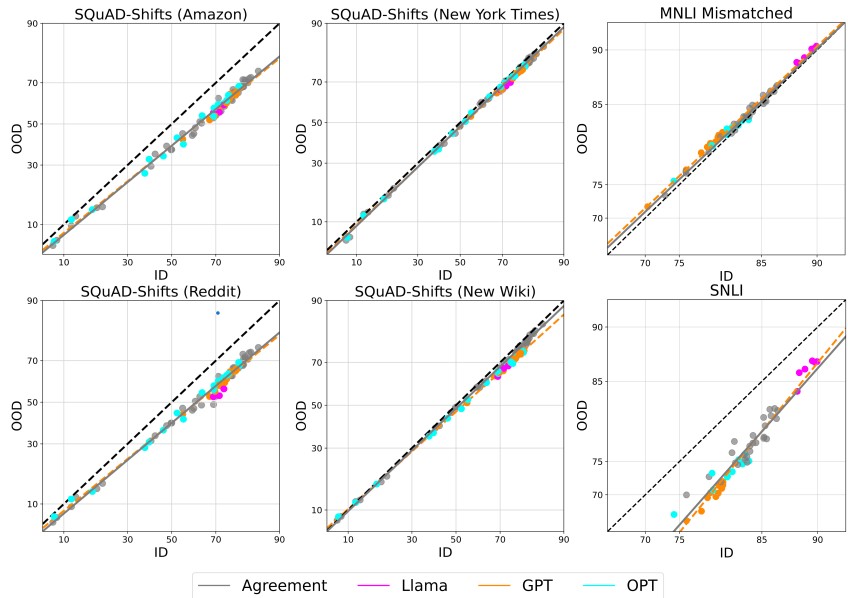

Figure 3: AGL can be observed between models finetuned from different base models (Llama, GPT, OPT) for the F1 score for question-answering shift (SQuAD to SQuAD-Shifts) and accuracy for text classification (MNLI-Matched to MNLI-Mismatched and SNLI). SQuAD-Shifts New Wiki, SQuAD-Shifts NYT, and MNLI Mismatched show little drop in OOD performance because the distribution shift is small compared to the corresponding ID dataset. Nonetheless, we observe that AGL holds regardless of the degree of distribution shift.

are only lightly fine-tuned or linearly probed, unlike deep ensembles trained from scratch. Diversity sources may behave differently in this circumstance.

## 4    Predicting OOD performance: multiple foundation models

Alternatively, we consider ensembling *multiple* base foundation models. First, ACL may not hold because the base models are heavily pretrained on different data corpuses. This may cause respective downstream models to have different ID versus OOD accuracy trends or "effective robustness" [17]. On vision tasks, for example, linear probing over CLIP, EfficientNet [50], ViT [15], and BYOL [20] observe varying robustness trends [43]. Second, even when ACL does hold, it is unclear whether the ensembles will also observe AGL. Here the problem is different from the single base model setting: any pair of foundation models finetuned from different base models may agree *too little*, or OOD agreement rate may vary across model pairs depending on the similarity of the pretraining corpus, breaking the linear correlation of agreement entirely. Yet, we observe that for language models and tasks, ensembles of finetuned FMs from a wide range of base models *observe both ACL and AGL*.

**Models**    We finetune models from OPT-125M, OPT-350M, OPT-1.3B [65], GPT2, GPT2-Medium, GPT2-Large, GPT2-XL [42], GPT-Neo-135M [5], Llama2-7B [53], Alpaca-7B [52], and Vicuna-7B [9]. We fully finetune OPT and GPT models and LoRA finetune Llama, Alpaca, and Vicuna. These models are pretrained on different mixtures of BookCorpus [66], Stories [55], PILE [18], CCNews v2 corpus, and PushShift.io Reddit [3]. Alpaca and Vicuna are instruction-finetuned over Llama2.

### 4.1    Results

We investigate the AGL behavior of an ensemble of foundation models finetuned from diverse base models in Figure 3 for question-answering. First note that base LLM models pretrained on different text corpora lead to finetuned models that lie on the *same linear trend in accuracy*. Unlike the different accuracy trends observed by different vision foundation models [42], we suspect that the pretraining datasets for the language models in our study observe much more homogeneity. Second, the ID

Table 3: The MAPE (%) of predicting OOD performance using AGL-based ALine and other baseline methods. We collect a diverse ensemble by randomizing the linear initialization and including multiple base models. *We filter out shifts with low correlation in agreement.

| OOD Dataset | ALine-D | ALine-S | Naive Agr | ATC | AC | DOC-Feat |
|---|---|---|---|---|---|---|
| CIFAR10C* | 5.44 | **4.73** | 17.39 | 6.90 | 11.49 | 11.91 |
| CIFAR10.1 v6 | **1.95** | 1.99 | 16.95 | 2.60 | 4.93 | 5.36 |
| CIFAR100C* | 7.17 | **6.79** | 17.66 | 8.18 | 17.58 | 14.96 |
| ImageNetC* | 15.03 | **14.17** | 32.27 | 15.90 | 22.83 | 13.42 |
| ImageNetV2 MatchFreq | 8.44 | 8.43 | 22.21 | **3.02** | 15.53 | 8.43 |
| fMoW-WILDS | 14.26 | **7.29** | 141.31 | 12.17 | 19.87 | 8.76 |
| OfficeHome-Art | 17.78 | **13.94** | 40.60 | 38.52 | 19.68 | 44.40 |
| OfficeHome-ClipArt | 14.04 | **11.70** | 33.44 | 33.81 | 14.97 | 28.77 |
| OfficeHome-Product | 14.64 | **11.78** | 39.88 | 84.18 | 63.05 | 75.44 |
| OfficeHome-Real | 12.65 | **10.18** | 36.20 | 27.72 | 21.85 | 28.85 |
| SQuAD-Shifts Reddit | 3.61 | **3.48** | 26.56 | 19.06 | 30.94 | 9.18 |
| SQuAD-Shifts Amazon | **3.61** | 4.93 | 26.46 | 24.35 | 34.93 | 7.31 |
| SQuAD-Shifts NYT | **1.64** | 1.75 | 23.46 | 4.01 | 25.96 | 2.80 |
| SQuAD-Shifts New Wiki | 6.33 | 6.58 | 25.24 | **5.18** | 25.96 | 7.50 |
| MNLI Mismatched | 0.55 | **0.41** | 0.55 | 12.00 | 0.63 | 0.51 |
| SNLI | 2.90 | **2.10** | 2.50 | 6.30 | 3.80 | 8.40 |

versus OOD agreement between pairs of models in this ensemble, including those between different base foundation models, is also strongly correlated and the slope and intercept closely matches that of accuracy. In other words, ensembles of different base models also observe AGL without any special regularization for ensemble diversity. The same holds for generative QA tasks (Appendix A.6.2).

## 5 Estimating OOD Accuracy using AGL in Diverse Ensembles

By constructing a diverse ensemble of foundation models, we can leverage AGL to extract precise estimates of model performances under distribution shift. We construct ensembles by collecting models trained from randomly-initialized heads (Section 3) and different base models (Section 4). For image classification, our model collection consists just linear models over CLIP representations. For text classification and question-answering, we include GPT, OPT, and Llama models individually finetuned from differently initialized heads. In Table 3, we compare the Mean Absolute Percentage Error (MAPE) of AGL-based prediction algorithms, ALine-S and ALine-D [2], to other baselines.

We compare against confidence based methods ATC [19], AC [24] and DOC-Feat [21] and Naive Agreement which directly uses agreement between model pairs [27, 35]. For confidence based methods, we first temperature scale the models using ID validation data, and pick the lower error rate from the estimations obtained with and without temperature scaling. However, there are several limitations when naively applying confidence baselines to estimate performance on question-answering, as they estimate classification accuracy. First, there is no easy analogous formulation of confidence baselines for the F1 score, so we estimate the exact-match score instead for fair comparison. On the other hand, AGL can predict performance across metrics accuracy, F1, and exact-match. Second, extractive question-answering is a joint classification task where models predict both the start and end token index of the answer span in the context. More details for how we calibrate baselines for this setting is provided in Appendix A.1.2.

Because ALine-S and ALine-D only provide estimation guarantees where the coefficient of determination $R^2$ of the linear fit in agreement is strong ([2]), we filter out datasets with low $R^2 \leq 0.95$. These shifts include iWildCam-WILDS, Camelyon-WILDS, and a few corruptions in CIFAR10C, CIFAR100C, and ImageNetC. We evaluate ALine-S/D for these failure cases in Appendix A.1.3. Across datasets with a high $R^2$ in agreement, ALine-S and ALine-D provide precise OOD performance estimates in finetuned FMs, surpassing other baselines by a large margin. This is noteworthy, especially for shifts where the agreement line is significantly off $y = x$, further lending to the utility of this method. Furthermore, they perform better on the question-answering task SQuAD, with the next best confidence method achieving as large as 20% higher error.

# 6    Limitations

Estimating the out-of-distribution performance of foundation models has rapidly grown in importance, especially as these models are increasingly deployed in real-world use cases. Our work focuses on a promising method to enable deployers to reduce the harm of machine learning systems when they encounter OOD inputs. However, deployers should be careful to not use AGL as the only signal for OOD performance. The correlation between agreement and accuracy is not guaranteed to hold for all distribution shifts, so other metrics should additionally be used to monitor model performance. In particular for foundation models, we observe that careful choices during fine-tuning is required to observe AGL. In fact, if different pretrained checkpoints are evaluated zero-shot, without any fine-tuning, AGL is not able to reliably predict the performance of FMs (Appendix A.3.5). Furthermore, while we studied AGL closely for a wide array of classification/QA benchmarks, there remains other important downstream tasks such as long-form generation that we leave for future study. We also do not provide any theoretical guarantees to back our empirical findings, such as the importance of random initialization, which we leave for future work.

# 7    Conclusion

We develop methods for extending AGL to foundation models to enable OOD performance prediction in this emerging paradigm. We find that utilizing AGL for performance estimation requires a careful tuning of ensemble diversity. Unlike the original paradigm of AGL, where models observed tens or hundreds of epochs of training on the ID dataset, we find that randomness in specific optimization choices, especially linear head initialization, is crucial for foundation models. In fact, in contrast to [2], we find that linear models can also observe AGL, specifically in the CLIP representation space, suggesting that AGL may not be a neural network specific phenomena. Our conclusion on AGL also sheds light on the robustness of foundation models. First, our experiments show that light finetuning alone can corrupt models to have diverse behaviors. Next, in contrast to vision models, where previous works show different forms of pretraining lead to different slopes in the linear correlations [43], we find that all the language models we evaluate, e.g., OPT, GPT2, GPT2-Neo, Alpaca, Llama, and Vicuna lie on the *same* accuracy line. This is particularly intriguing because it goes against the common wisdom that the pretraining data influences the models' "effective robustness". We leave these questions for future analysis.

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

# A Appendix

**Contents**

### A.1 Details Regarding OOD Performance Estimation Baselines

#### A.1.1 AGL-Based Estimation Methods: ALine-S/D

ALine algorithms are the AGL-based performance estimation methods proposed in [2]. When the AGL phenomenon occurs, i.e., models observe a strong linear correlation in both ID versus OOD agreement and accuracy with matching slopes and biases, algorithms ALine-S and ALine-D effectively apply the linear transformation calculated using agreements to map the ID performances to OOD performance estimates. We describe the algorithms in more detail below.

**AGL** Provided a collection of models $\mathcal{F} = \{f_1, f_2, ..., f_n\}$, AGL suggests that ID versus OOD accuracy observe a strong linear correlation if and only if ID versus OOD agreement observes a strong linear correlation and when they do, the slopes and biases match: $\forall f_i, f_j \in \mathcal{F}$ where $i \neq j$

$$\Phi^{-1}(\text{Acc}_{\text{OOD}}(f_i)) = a \cdot \Phi^{-1}(\text{Acc}_{\text{ID}}(f_i)) + b$$
$$\Updownarrow \qquad\qquad (3)$$
$$\Phi^{-1}(\text{Agr}_{\text{OOD}}(f_i, f_j)) = a \cdot \Phi^{-1}(\text{Agr}_{\text{ID}}(f_i, f_j)) + b$$

$\Phi^{-1}$ is the probit transform used to induce a better linear fit as used in [2] and [39]. Provided access to $\text{Acc}_{\text{ID}}(f_i), \text{Agr}_{\text{ID}}(f_i, f_j), \text{Agr}_{\text{OOD}}(f_i, f_j) \forall i, j$, we'd like to estimate $\text{Acc}_{\text{OOD}}(f_i)$ for all $f_i \in \mathcal{F}$.

**ALine-S** The algorithm ALine-S simply estimates the the slope $a$ and bias $b$ of accuracy by computing the linear fit of agreement.

$$\hat{a}, \hat{b} = \arg\min_{a, b \in \mathbb{R}} \sum_{i \neq j} \left( \Phi^{-1}(\hat{\text{Agr}}_{\text{OOD}}(f_i, f_j)) - a \cdot \Phi^{-1}(\hat{\text{Agr}}_{\text{ID}}(f_i, f_j)) - b \right)^2 \qquad (4)$$

With $\hat{a}$ and $\hat{b}$, we estimate $\text{Acc}_{\text{OOD}}(f_i) \approx \hat{a} \cdot \text{Acc}_{\text{ID}}(f_i) + \hat{b}$. This method is called Aline-S.

**ALine-D** This method instead constructs the following system of linear equations. Provided the relation in Equation 2, one can derive that for any $f_i, f_j \in \mathcal{F}$,

$$\frac{1}{2} \left( \Phi^{-1}(\text{Acc}_{\text{OOD}}(f_i)) + \Phi^{-1}(\text{Acc}_{\text{OOD}}(f_j)) \right)$$
$$\approx \Phi^{-1}(\text{Agr}_{\text{OOD}}(f_i, f_j)) + \hat{a} \cdot \left( \frac{1}{2}\Phi^{-1}(\text{Acc}_{\text{ID}}(f_i)) + \frac{1}{2}\Phi^{-1}(\text{Acc}_{\text{ID}}(f_j)) - \Phi^{-1}(\text{Agr}_{\text{ID}}(f_i, f_j)) \right)$$
$$(5)$$

Treating $\text{Acc}_{\text{OOD}}(f_i) \forall i$ as unknown variables, note that the right hand side is known and we can construct a linear system of equations using all $\binom{n}{2}$ pairs of models. The algorithm employs least squares to solve this approximate system of linear equations.

### A.1.2 Temperature Scaling for Confidence-Based Estimation Methods

We compare ALine against confidence-based methods ATC [19], AC [24], and Doc-Feat [21]. These methods notably perform better after calibrating the models ID by temperature scaling.

**Classification**   For classification tasks, we optimize a temperature $T$ for each model $f$ on the cross-entropy loss over the in-distribution validation data.

$$\min_T \sum_{x,y} \mathsf{CE}(\sigma(f(x)\exp(T)), y) \tag{6}$$

where $\sigma(\cdot)$ is the softmax.

**Question-Answering**   For extractive question-answering tasks, the model has to predict two labels – the start and end token index $y = [y_s, y_e]$ of the context span that answers the question. For each model, we attach a span prediction head $v = [v_s, v_e] \in \mathbb{R}^{d\times 2}$ on top of the base $\mathsf{B}_\theta(x) \in \mathbb{R}^{d\times N}$ where $N$ is the token length of $x$. $s(x) = v_s^\top \mathsf{B}_\theta(x)$ and $e(x) = v_e^\top \mathsf{B}_\theta(x)$ predict the start and end token index, respectively.

We're interested in evaluating question-answering models on the exact match (EM) objective,

$$\mathsf{EM}(\hat{y}, y) = \mathbb{1}\left[\hat{y}_s = y_s\right] \cdot \mathbb{1}\left[\hat{y}_e = y_e\right] \tag{7}$$

EM treats question-answering as a classification problem over $N \times N$ choices of start and end index pairs. This allows us to utilize confidence-based methods that are designed for classification tasks. We can calculate the model confidence for index pair $[i, j]$ as $\sigma(s(x))_i \cdot \sigma(e(x))_j$.

We jointly optimize a separate temperature for the start and end logits $T_s$ and $T_e$ and we minimize the cross-entropy loss over the in-distribution validation data.

$$\min_T \sum_{x,y} \mathsf{CE}(\sigma(s(x)\exp(T_s))\sigma(e(x)\exp(T_e))^\top, y) \tag{8}$$

### A.1.3 Comparison to ProjNorm

In this section, we present a comparison with ProjNorm [63], a method that yields a score which is shown to be correlated with the OOD performance of the model. We study the same setting as presented in Section 4 where we estimate OOD performance of foundation models pretrained on different text corpora. Unlike AGL, ProjNorm doesn't provide an estimate of the OOD performance hence we compare the linear correlation between the predicted value and OOD performance. From Table 4 it can be seen that estimates from ALine-D are more strongly correlated with OOD performance than ProjNorm.

Table 4: Correlation coefficient between OOD Accuracy and Prediction for ALine-D and ProjNorm

| Method | SQuAD-Shifts Amazon | SQuAD-Shifts Reddit |
|---|---|---|
| ALine-D | **0.98** | **0.98** |
| ProjNorm | 0.64 | 0.79 |

### A.1.4 Failure Datasets with Low Linear Correlation

In our comparison with baselines in Section 5, we filter out datasets with a low correlation coefficient $\leq 0.95$ in ID vs OOD agreement. When the linear correlation of agreement is weak, AGL tells us that the correlation is also low for accuracy, and AGL-based methods are not guaranteed to be reliable in such circumstances. We provide ID vs OOD accuracy and agreement scatter plots for all datasets in Appendix A.3.2.

Below, we separately provide the comparison with baselines for the excluded datasets. We generally find that the baseline ATC [19] is significantly better in circumstances where AGL-based methods are unreliable.

Table 5: The MAPE (%) of predicting OOD performance using AGL-based ALine and other baseline methods. We collect a diverse ensemble by randomizing the linear initialization and including multiple base models.

| OOD Dataset | Agreement $R^2$ | ALine-D | ALine-S | Naive Agr | ATC | AC | DOC-Feat |
|---|---|---|---|---|---|---|---|
| CIFAR10C Gaussian Noise | 0.85 | 65.59 | 56.67 | **41.85** | 42.77 | 74.60 | 75.40 |
| CIFAR10C Glass Blur | 0.89 | 44.24 | 44.97 | **31.24** | 33.58 | 79.45 | 80.36 |
| CIFAR10C Shot Noise | 0.89 | 45.02 | 37.60 | **27.70** | 28.25 | 49.23 | 49.90 |
| CIFAR10C Speckle Noise | 0.90 | 36.67 | 30.13 | 23.45 | **22.72** | 43.46 | 44.12 |
| CIFAR100C Gaussian Noise | 0.93 | 34.98 | 32.21 | 28.88 | **21.18** | 69.04 | 63.81 |
| CIFAR100C Glass Blur | 0.93 | 66.03 | 63.45 | 42.41 | **25.47** | 109.73 | 103.70 |
| ImageNetC Gaussian Noise | 0.89 | 50.10 | 47.26 | 73.47 | **13.21** | 54.86 | 42.66 |
| ImageNetC Glass Blur | 0.87 | 74.20 | 71.84 | 100.32 | **15.97** | 74.12 | 59.60 |
| ImageNetC Impulse Noise | 0.88 | 62.86 | 59.68 | 87.83 | **16.54** | 63.23 | 49.99 |
| ImageNetC Shot Noise | 0.88 | 53.89 | 51.12 | 78.37 | **14.66** | 57.47 | 44.58 |
| iWildCam-WILDS | 0.85 | **22.05** | 25.29 | 46.42 | 37.25 | 57.31 | 69.58 |
| Camelyon17-WILDS | 0.59 | 10.14 | **6.44** | 13.26 | 6.46 | 8.76 | 8.90 |

## A.2 Finetuning Hyperparameters

We state here the hyperparameters used to finetune the models for diversity experiments reported in Section 3.

### A.2.1 Linear Probing over CLIP for Vision Tasks

We train all linear probes using SGD. Models are trained for different timesteps to achieve an even distribution of ID accuracies.

Table 6: CLIP Linear Probing

| Dataset | Hyperparameters |
|---------|-----------------|
| CIFAR10 | Learning Rate: $5 \times 10^{-4}$
Batch Size: 1028 |
| CIFAR100 | Learning Rate: $1 \times 10^{-3}$
Batch Size: 1028 |
| ImageNet | Learning Rate: $1 \times 10^{-1}$
Batch Size: 1028 |
| OfficeHome | Learning Rate: $1 \times 10^{-3}$
Batch Size: 200 |
| FMoW-WILDS | Learning Rate: $1 \times 10^{-3}$
Batch Size: 200 |
| iWildCam-WILDS | Learning Rate: $1 \times 10^{-3}$
Batch Size: 200 |
| Camelyon17-WILDS | Learning Rate: $1 \times 10^{-3}$
Batch Size: 200 |

### A.2.2 Full Finetuning GPT, OPT, BERT on Language Tasks

We use AdamW [34] to full finetune language models. We keep the learning rate small. Models are trained for different timesteps to achieve an even distribution of ID accuracies.

Table 7: Full Finetuning GPT2-Medium on SQuAD

| Source of Diversity | Hyperparameters |
|---------------------|-----------------|
| Initialization + Ordering | Learning rate: $2 \times 10^{-7}$
Weight Decay: $1 \times 10^{-5}$
Batch Size: 32
Max Epochs: 20 |
| Subsetting (10, 30 % of data) | Learning rate: $6, 4 \times 10^{-7}$
Weight Decay: $1 \times 10^{-5}$
Batch Size:
Max Epochs: 20 |

Table 8: Full Finetuning OPT-125M on SQuAD

| Source of Diversity | Hyperparameters |
|---|---|
| Initialization + Ordering | Learning rate: $4 \times 10^{-7}$
Weight Decay: $1 \times 10^{-5}$
Batch Size: 32
Max Epochs: 20 |
| Subsetting (10, 30, 50 % of data) | Learning rate: $40, 12, 8 \times 10^{-7}$
Weight Decay: $1 \times 10^{-5}$
Batch Size: 32
Max Epochs: 20 |

Table 9: Full Finetuning BERT-Uncased on SQuAD

| Source of Diversity | Hyperparameters |
|---|---|
| Initialization + Ordering | Learning rate: $2 \times 10^{-7}$
Weight Decay: $1 \times 10^{-5}$
Batch Size: 32
Max Epochs: 20 |
| Subsetting (10, 30, 50 % of data) | Learning rate: $20, 6, 4 \times 10^{-7}$
Weight Decay: $1 \times 10^{-5}$
Batch Size: 32
Max Epochs: 20 |

Table 10: Full Finetuning GPT2-Medium on MNLI

| Source of Diversity | Hyperparameters |
|---|---|
| Initialization + Ordering | Learning rate: $5 \times 10^{-4}$
Weight Decay: $1 \times 10^{-5}$
Batch Size: 128
Max Epochs: 10 |
| Subsetting (10% of data) | Learning rate: $5 \times 10^{-3}$
Weight Decay: $1 \times 10^{-5}$
Batch Size: 128
Max Epochs: 10 |

Table 11: Full Finetuning OPT-125M on MNLI

| Source of Diversity | OPT-125M | |
| | Varied | Fixed |
|---|---|---|
| Initialization + Ordering | Learning rate: $1 \times 10^{-3}$
Weight Decay: $1 \times 10^{-5}$
Batch Size: 128
Max Epochs: 10 | |
| Subsetting (10% of data) | Learning rate: $1 \times 10^{-2}$
Weight Decay: $1 \times 10^{-5}$
Batch Size: 128
Max Epochs: 10 | |

## A.3 More Experiments on the Effect of Diversity Source with CLIP

### A.3.1 OfficeHome Linear Probing Diversity Experiments

In Section 3.1, we examine how diversity source impacts whether AGL is observed in linear probed CLIP models from CIFAR10 to CIFAR10C. Here, we perform the same experiment on Office-Home [56], which consists of 4 domains or image styles ("Art", "Clip Art", "Product", and "Real World") for 65 common objects. We train models on one domain and treat the remaining three domains as OOD. Similarly, only Random Initialization yields AGL or matching slopes in accuracy and agreement, and as a result, the corresponding MAPE of estimating the OOD performance of this diverse ensemble is the smallest.

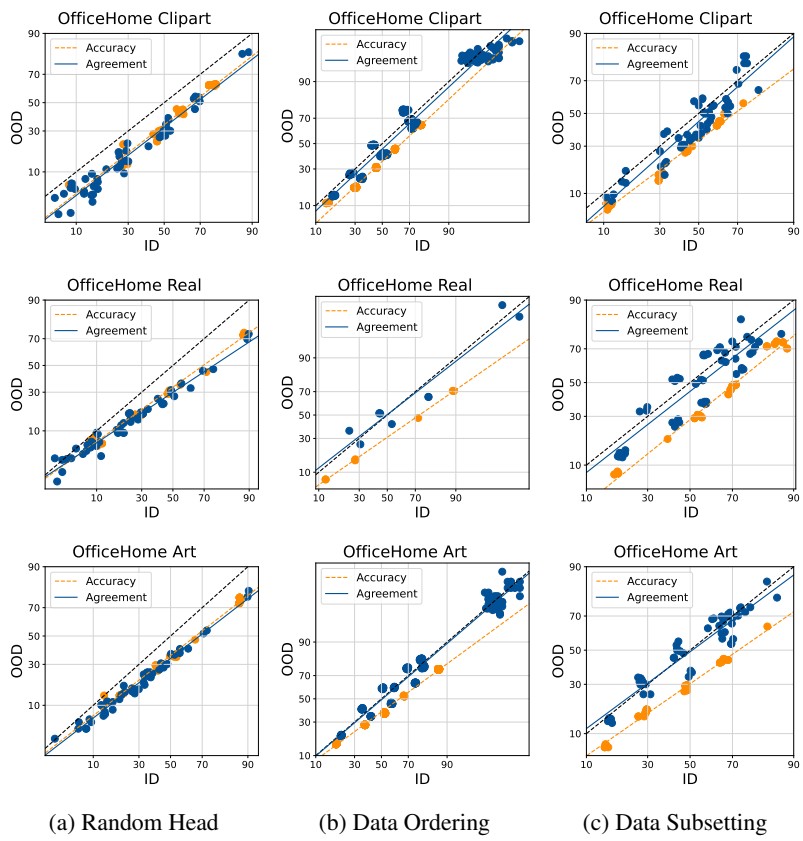

(a) Random Head      (b) Data Ordering      (c) Data Subsetting

Figure 4: ID vs OOD accuracy and agreement of linear probed CLIP models on OfficeHome Art (top row), Product (middle row), and Real World (bottom row). The figure title is the OOD domain.

Table 12: The ALine-S MAPE(%) for ensembles trained on each domain of OfficeHome.

| OfficeHome ID domain | Source of Diversity | OOD Estimation MAPE(%) |
|---|---|---|
| Art | Random Linear Heads | **14.09** |
| | Data Ordering | 20.67 |
| | Data Subsetting | 120.85 |
| ClipArt | Random Linear Heads | **12.23** |
| | Data Ordering | 28.45 |
| | Data Subsetting | 78.49 |
| Product | Random Linear Heads | **13.63** |
| | Data Ordering | 110.45 |
| | Data Subsetting | 92.97 |
| Real | Random Linear Heads | **9.95** |
| | Data Ordering | 33.36 |
| | Data Subsetting | 3078 |

### A.3.2 AGL appears in Random Head CLIP Ensembles across Datasets

We report the strength of AGL in linear probed CLIP models with randomly initialized linear heads across all datasets discussed in Section 3.2.

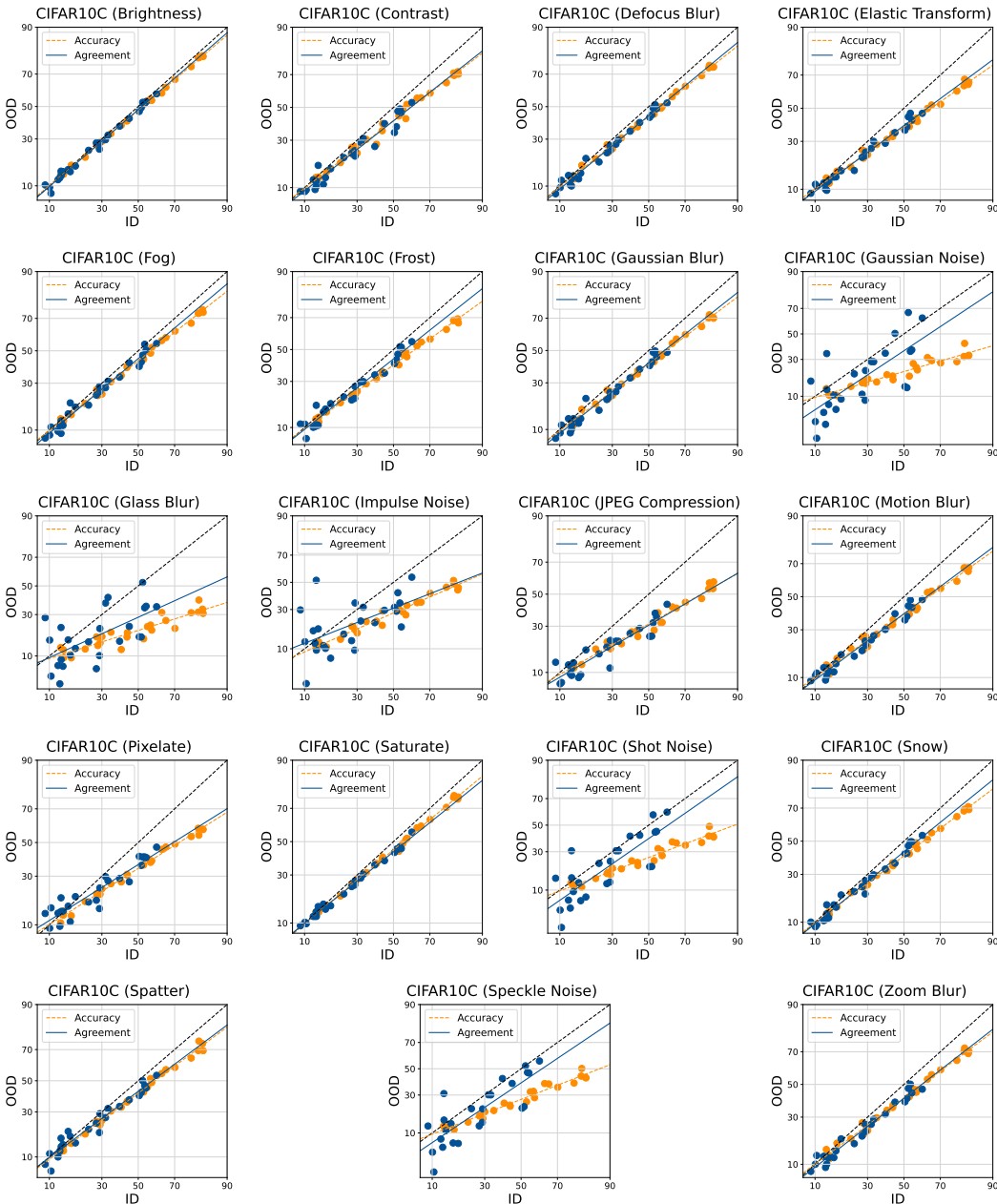

Figure 5: AGL and ACL for all C10C shifts with random head initialization finetuning.

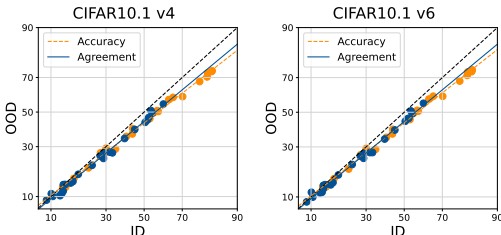

Figure 6: AGL and ACL for the C10.1 shifts with random head initialization finetuning.

Figure 7: AGL and ACL for the C100C shifts with random head initialization finetuning.

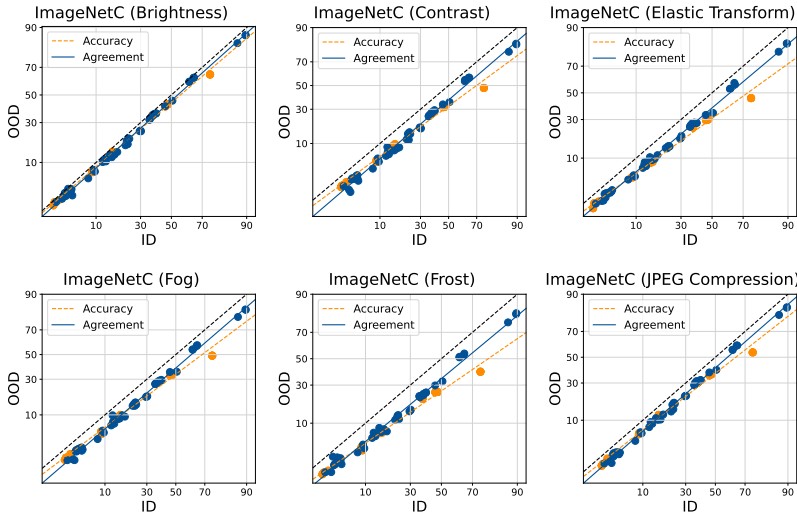

Figure 8: AGL and ACL for the ImageNetC shifts with random head initialization finetuning.

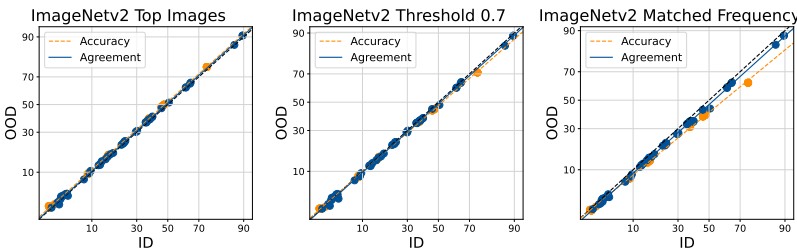

Figure 9: AGL and ACL for the ImageNet V2 shifts with random head initialization finetuning.

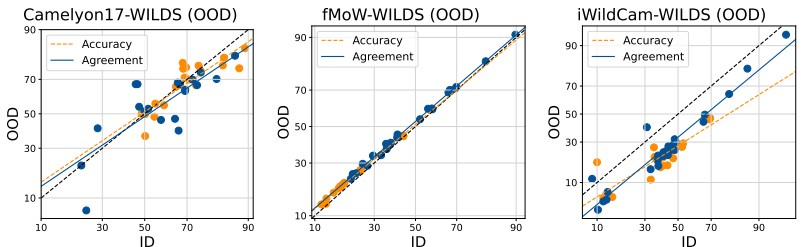

Figure 10: AGL and ACL for 3 benchmarks from the WILDS dataset with random head initialization finetuning.

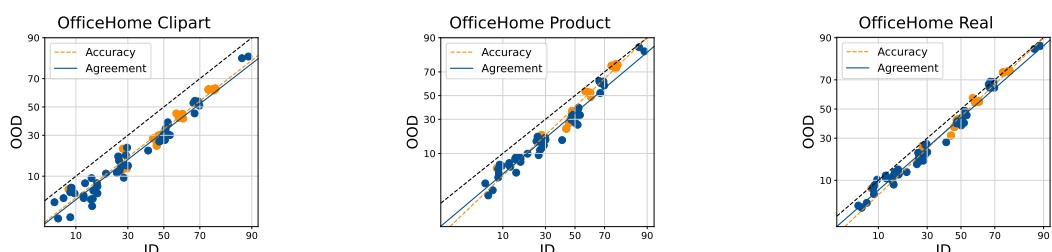

Figure 11: AGL and ACL for the OfficeHome ClipArt, Product, Real shifts with random head initialization finetuning over OfficeHome Art.

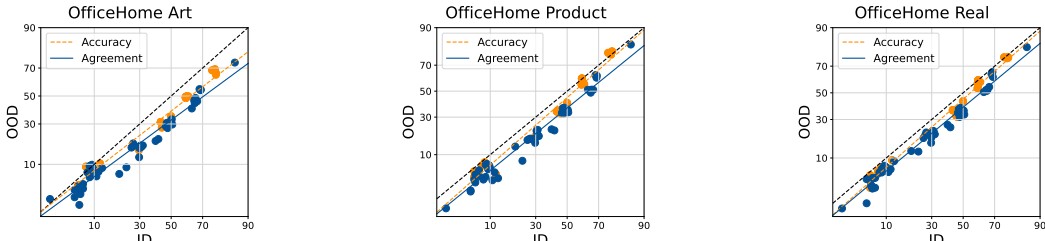

Figure 12: ID vs OOD accuracy and agreement with OH ClipArt ID

Figure 13: AGL and ACL for the OfficeHome Art, Product, Real shifts with random head initialization finetuning over OfficeHome ClipArt.

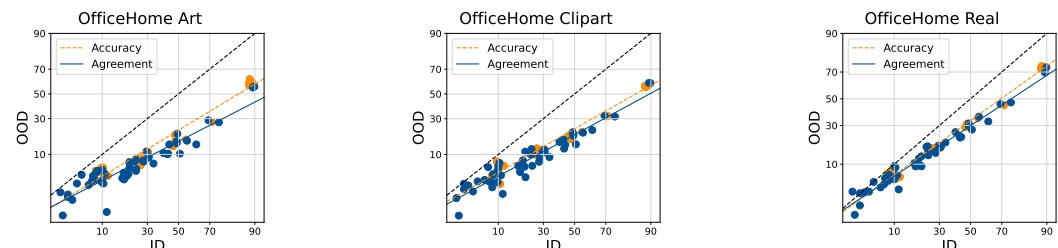

Figure 14: AGL and ACL for the OfficeHome ClipArt, Art, Real shifts with random head initialization finetuning over OfficeHome Product.

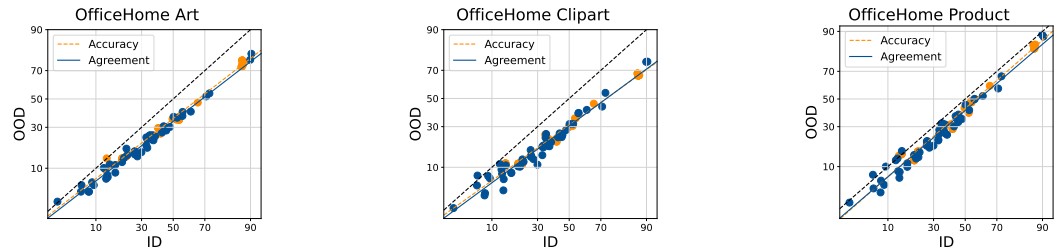

Figure 15: AGL and ACL for the OfficeHome Art, ClipArt, Product shifts with random head initialization finetuning over OfficeHome Real.

### A.3.3 Diversity Matters in Full Finetuned CLIP Ensembles

We show that light full finetuning over CLIP also observes similar effects of diversity source on the strength of AGL. We verify this on shifts from CIFAR10 to CIFAR10C.

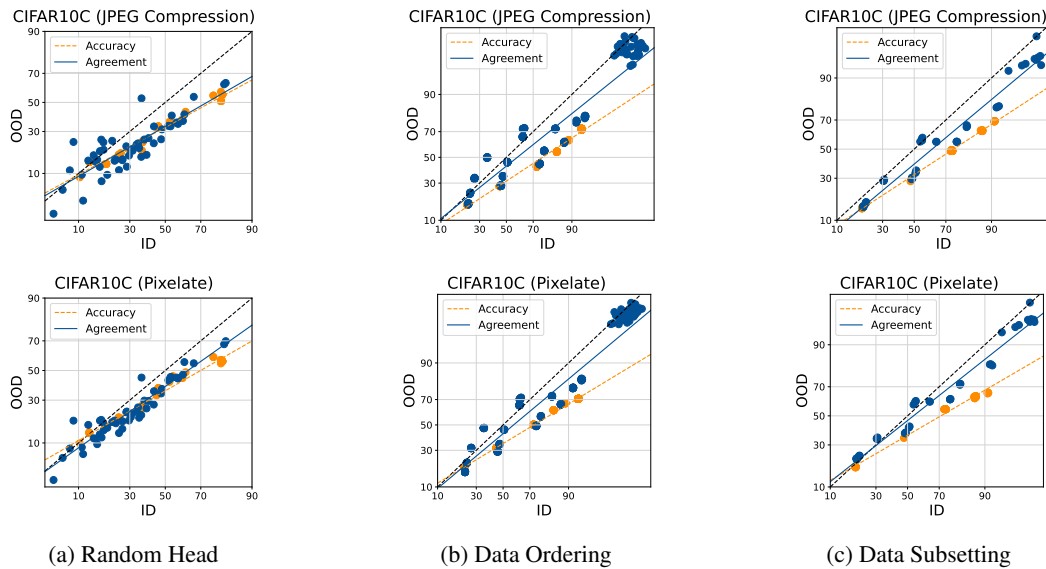

(a) Random Head        (b) Data Ordering        (c) Data Subsetting

Figure 16: ID vs OOD accuracy and agreement of full finetuned CLIP models on shift from CIFAR10 to CIFAR10C "JPEG Compression" (top row) and "Pixelate" (bottom row) shifts. Only Random Initialization (Column a) yields AGL or matching slopes in accuracy and agreement.

Table 13: The average MAPE(%) of ALine estimates of OOD performance of full finetuned CLIP models across all 19 CIFAR10C shifts. As can be seen from Figure 4, only ensembles with diverse random initialization consistently results in smallest MAPE values.

| Source of Diversity | CIFAR10C MAPE(%) |
| --- | --- |
| Random Linear Heads | **29.37** |
| Data Ordering | 69.98 |
| Data Subsetting | 33.57 |

### A.3.4 Any diverse ensemble display AGL in models heavily trained from scratch

On the other hand, we demonstrate that in models heavily trained from scratch, AGL can be observed irrespective of the diversity source. Consistent with the models in [2], we train ResNet18 on CIFAR10 from scratch, varying the different sources of randomness. These models are trained heavily with SGD with learning rate of $1 \times 10^{-2}$, batch size 128, and weight decay of $1 \times 10^{-5}$ for up to 200 epochs. We do not use any data augmentation.

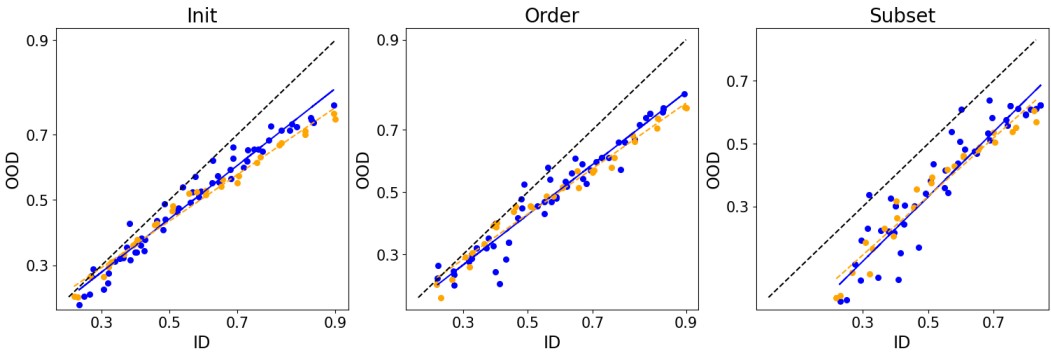

Figure 17: Effect of Diversity Source on ResNet18 from CIFAR10 to CIFAR10C-Snow

### A.3.5 AGL in FMs under Zero-Shot and Few-Shot Settings

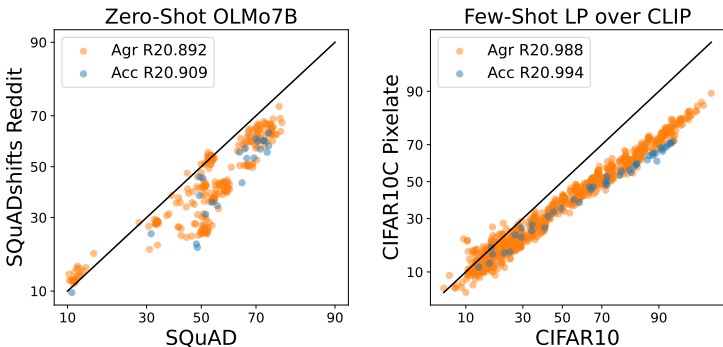

Figure 18: Zero-Shot and Few-Shot Settings. (Left) We plot the ID versus OOD zero-shot performances of OLMo7B checkpoints. We see that the linear correlation is weak in both F1 and F1-Agreement. This means that the effective robustness of base models vary widely during pretraining. (Right) We train linear probes over CLIP embeddings on few-shot CIFAR10 (10 examples per class) with random initialization. We similarly observe AGL and ACL with random initialization.

### A.3.6 Effect of Training Dataset Size

To rid of any confounding factors from data subsetting observing a smaller amount of data, we evaluate all randomness sources on different training dataset sizes. We track the effect of diversity in random initialization, data ordering, and data subsetting for different portions of the training data ($100\%$, $50\%$, $30\%$, $10\%$). For each percentage $x\%$, Random Initialization and Data Ordering ensembles are trained on the same randomly sampled $x\%$ proportion of the data while each model in the Data Subsetting ensemble observe different randomly sampled $x\%$ portions

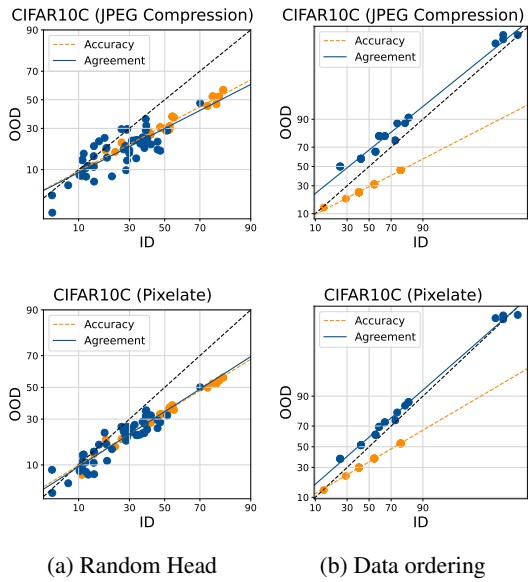

     (a) Random Head       (b) Data ordering

Figure 19: ID vs OOD accuracy and agreement of linear probe CLIP models finetuned on 100% of CIFAR10 training data evaluated on the CIFAR10C "JPEG Compression" (top row) and "Pixelate" (bottom row) shifts

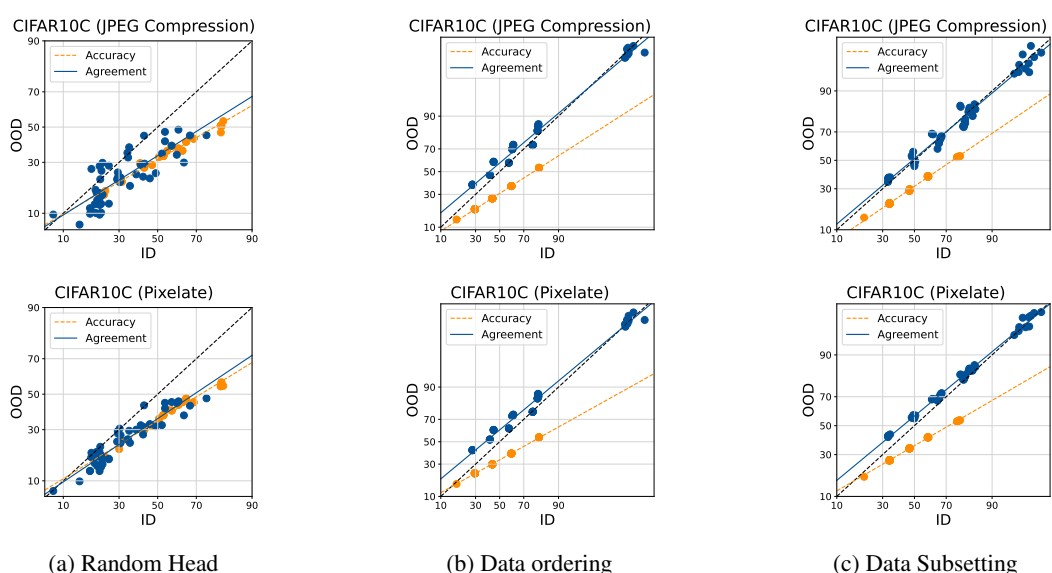

    (a) Random Head             (b) Data ordering             (c) Data Subsetting

Figure 20: ID vs OOD accuracy and agreement of linear probe CLIP models finetuned on 50% of CIFAR10 training data evaluated on the CIFAR10C "JPEG Compression" (top) and "Pixelate" (bottom) shifts

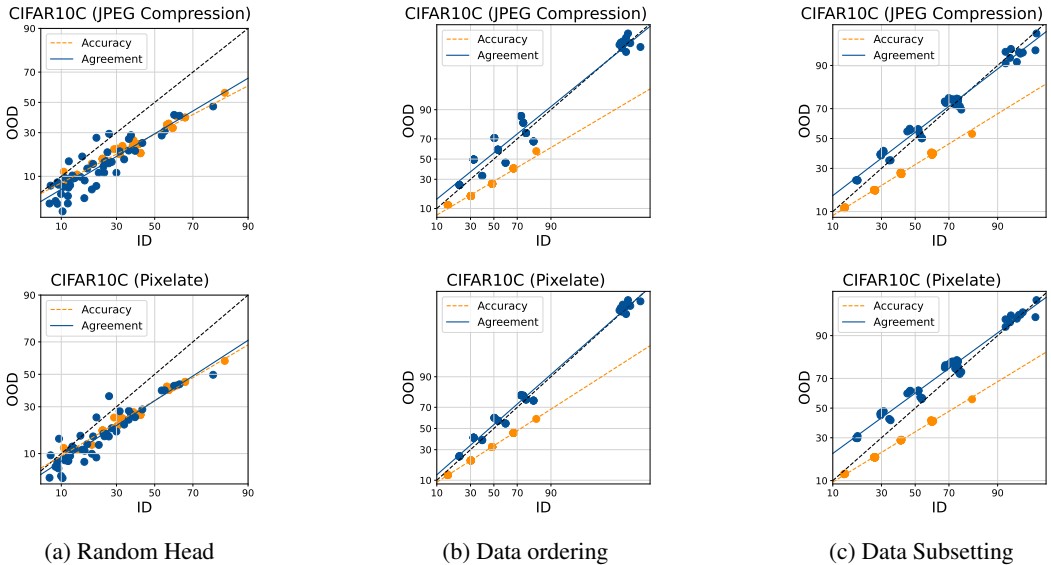

(a) Random Head       (b) Data ordering       (c) Data Subsetting

Figure 21: ID vs OOD accuracy and agreement of linear probe CLIP models finetuned on 30% of the CIFAR10 training data evaluated on the CIFAR10C "JPEG Compression" (top) and "Pixelate" (bottom) shifts

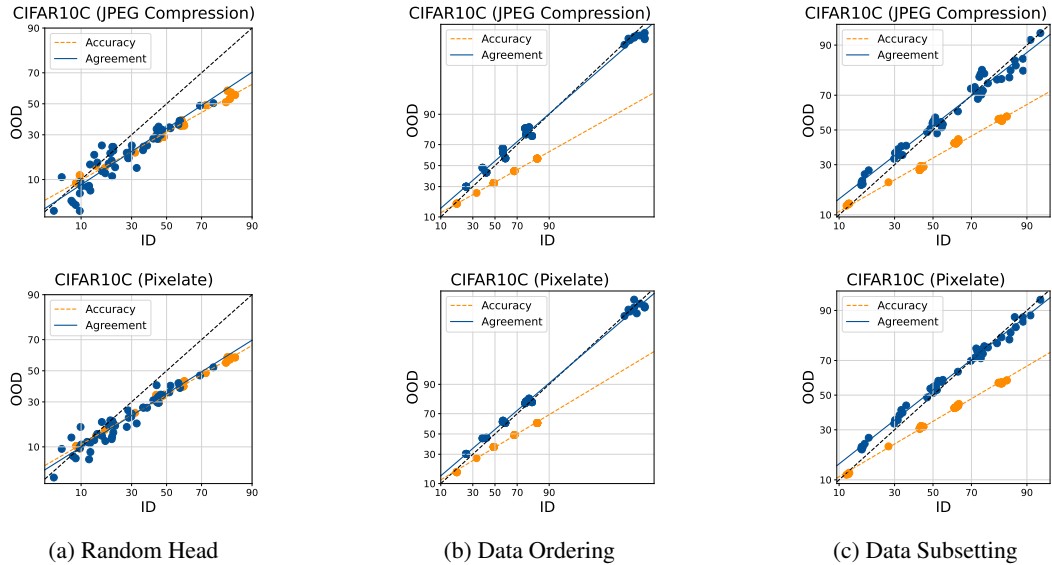

(a) Random Head       (b) Data Ordering       (c) Data Subsetting

Figure 22: ID vs OOD accuracy and agreement of linear probe CLIP models finetuned on 10% of the CIFAR10 training data and evaluated on the CIFAR10C "JPEG Compression" (top) and "Pixelate" (bottom) shifts

## A.4 More Experiments on the Effect of Diversity Source for Extractive Question Answering

### A.4.1 Single-Base Diversity Experiments Using OPT and BERT

In Section 3.2, we report the effect of diversity source in full finetuned GPT2-Medium models for the extractive QA task SQuAD to SQuAD-Shifts. In this section, we also provide the same experiments on OPT-125M and BERT. As observed in GPT, using Random Heads yields the strongest AGL behavior and achieves the smallest ALine-D MAPE.

In Table 14 and 15, we report the MAPE of OOD performance estimation of models trained with Random Initialization and Data Ordering using 100% of training data and Data Subsetting with 10% of training data.

Table 14: The average MAPE (%) of ALine-D OOD performance estimates of OPT-125M models full finetuned on SQuAD.

| Source of Diversity | SQuAD-Shifts Amazon | SQuAD-Shifts Reddit |
|---|---|---|
| Random Head | **6.54** | **5.43** |
| Data Ordering | 11.37 | 8.70 |
| Data Subsetting | 11.15 | 9.65 |

Table 15: The average MAPE (%) of ALine-D OOD performance estimates of BERT models full finetuned on SQuAD.

| Source of Diversity | SQuAD-Shifts Amazon | SQuAD-Shifts Reddit |
|---|---|---|
| Random Head | **14.70** | **8.62** |
| Data Ordering | 16.55 | 9.16 |
| Data Subsetting | 22.13 | 18.64 |

### A.4.2 Diversity in Full Finetuned GPT2 for Different Data Portions

We track the effect of diversity in random initialization, data ordering, and data subsetting for different portions of the training data ($100\%$, $50\%$, $30\%$, $10\%$). For each percentage $x\%$, Random Initialization and Data Ordering ensembles are trained on the same randomly sampled $x\%$ proportion of the data while each model in the Data Subsetting ensemble observe different randomly sampled $x\%$ portions.

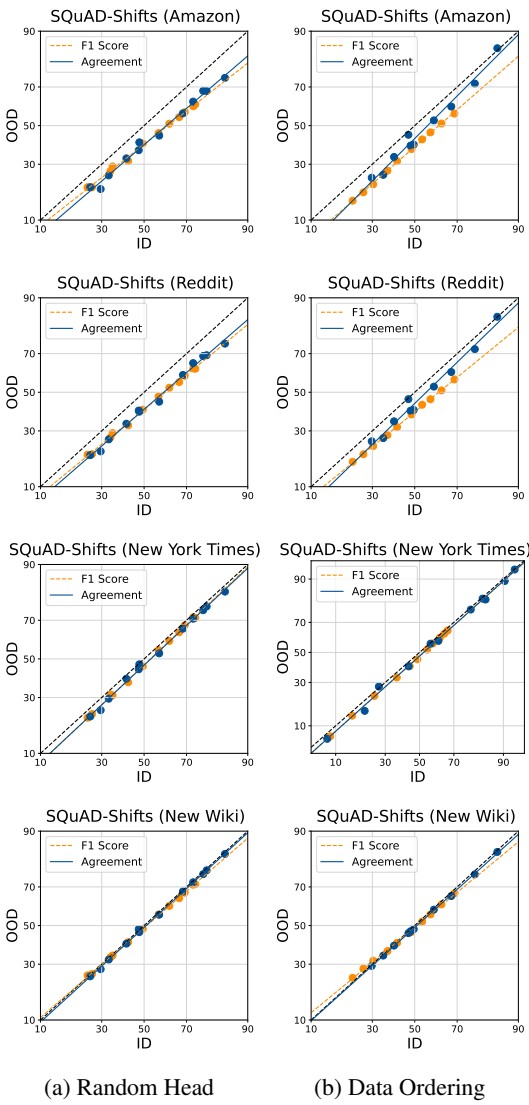

(a) Random Head          (b) Data Ordering

Figure 23: ID vs OOD accuracy and agreement of models finetuned on SQuAD from a single pretrained GPT2 model with 100% of the training data

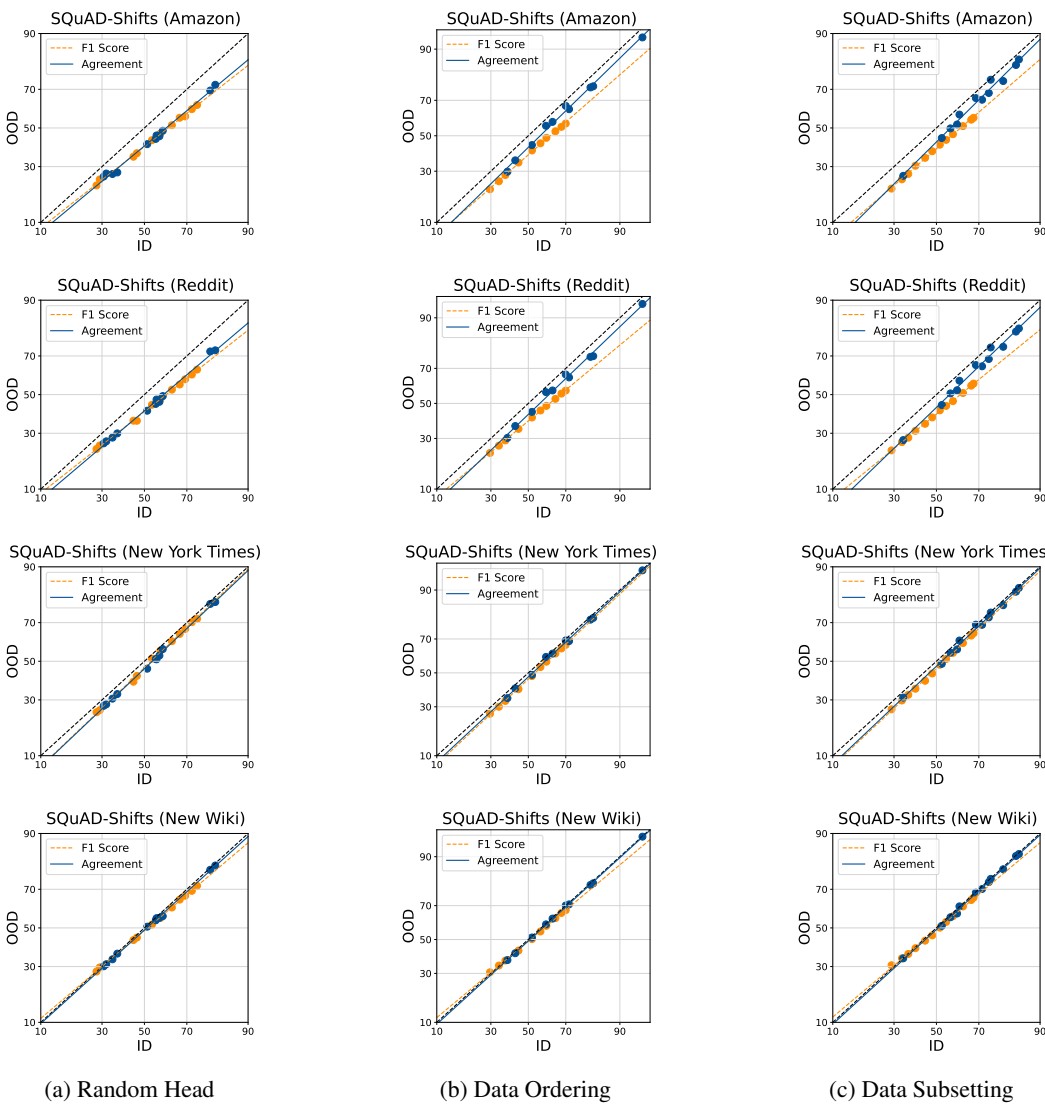

(a) Random Head        (b) Data Ordering        (c) Data Subsetting

Figure 24: ID vs OOD accuracy and agreement of models finetuned on SQuAD from a single pretrained GPT2 model with 50% of the training data

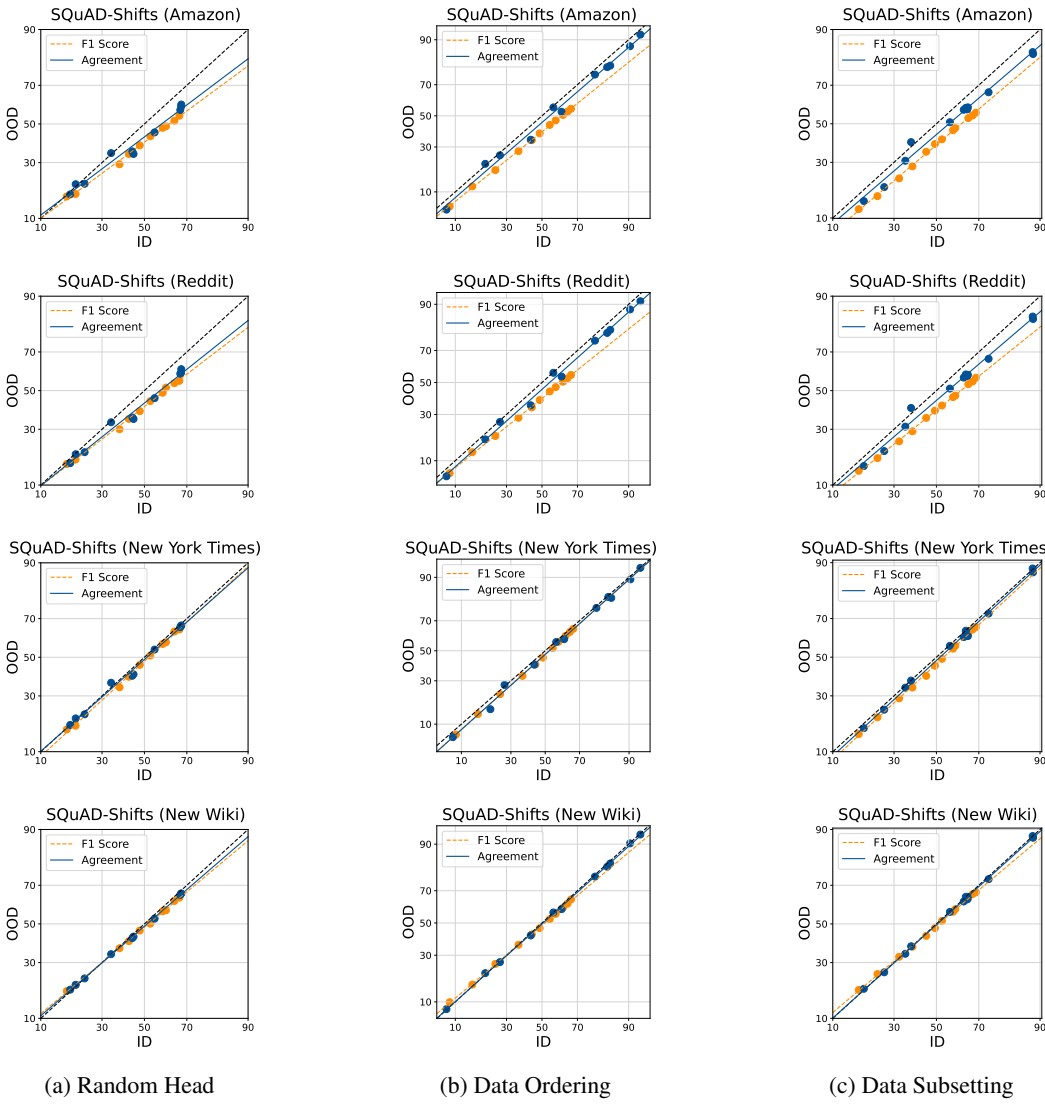

(a) Random Head     (b) Data Ordering     (c) Data Subsetting

Figure 25: ID vs OOD accuracy and agreement of models finetuned on SQuAD from a single pretrained GPT2 model with 10% of the training data

### A.4.3 Diversity in Full Finetuned OPT for Different Data Portions

We track the effect of diversity in random initialization, data ordering, and data subsetting for different portions of the training data ($100\%$, $50\%$, $30\%$, $10\%$). For each percentage $x\%$, Random Initialization and Data Ordering ensembles are trained on the same randomly sampled $x\%$ proportion of the data while each model in the Data Subsetting ensemble observe different randomly sampled $x\%$ portions.

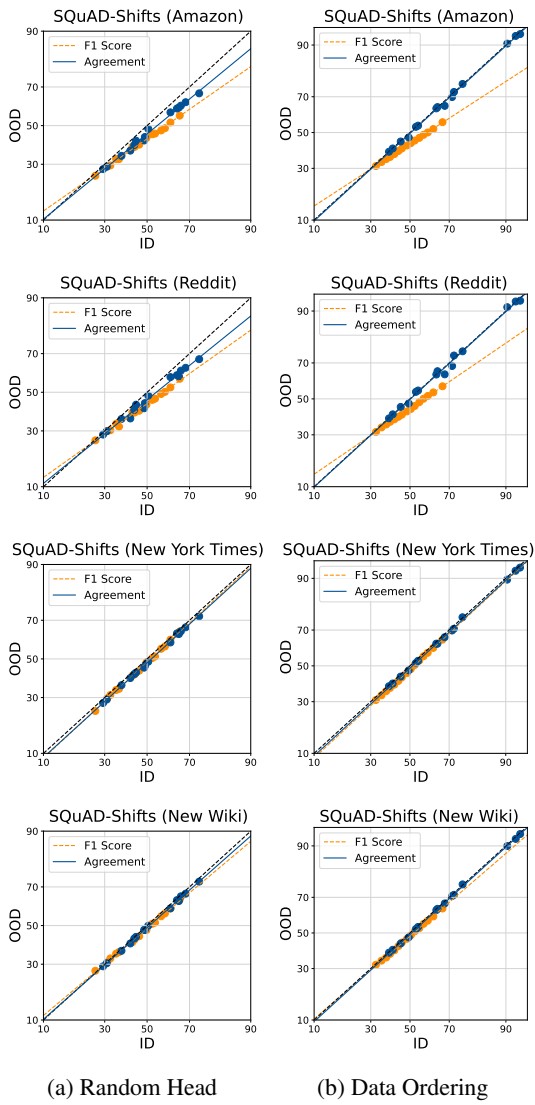

(a) Random Head      (b) Data Ordering

Figure 26: ID vs OOD accuracy and agreement of models finetuned on SQuAD from a single pretrained OPT-125m with $100\%$ of the training data

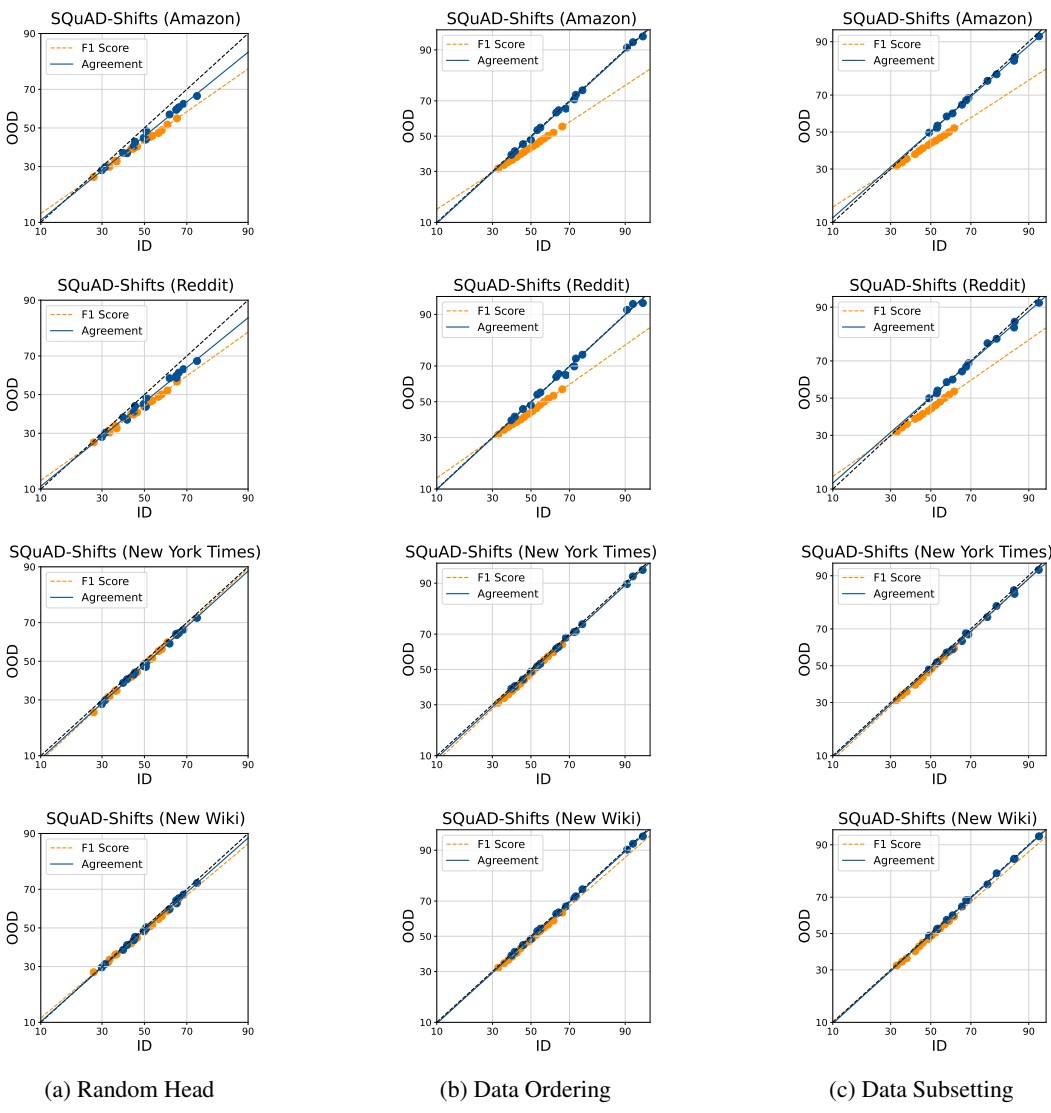

Figure 27: ID vs OOD accuracy and agreement of models finetuned on SQuAD from a single pretrained OPT-125m with 50% of the training data

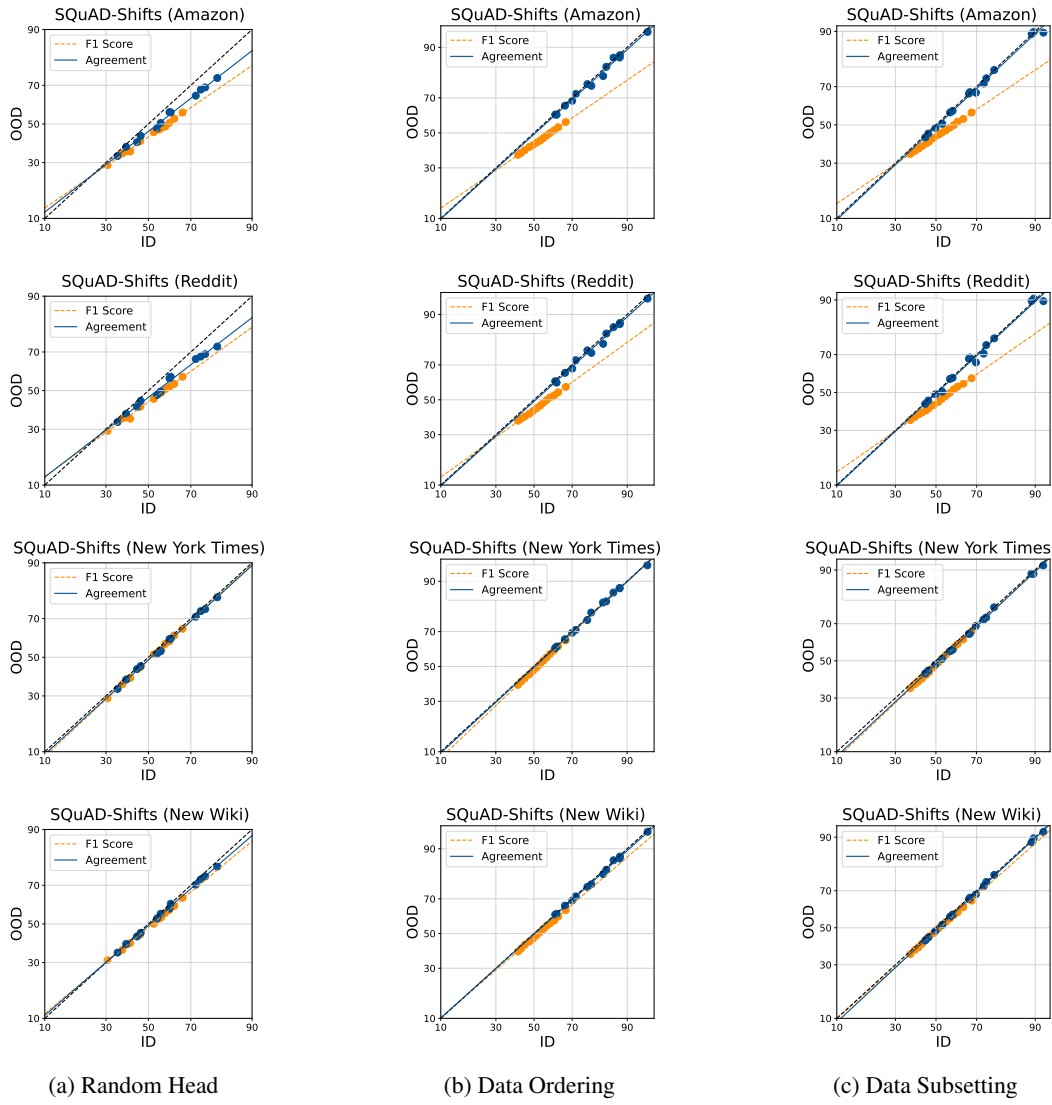

(a) Random Head          (b) Data Ordering          (c) Data Subsetting

Figure 28: ID vs OOD accuracy and agreement of models finetuned on SQuAD from a single pretrained OPT-125m with 30% of the training data

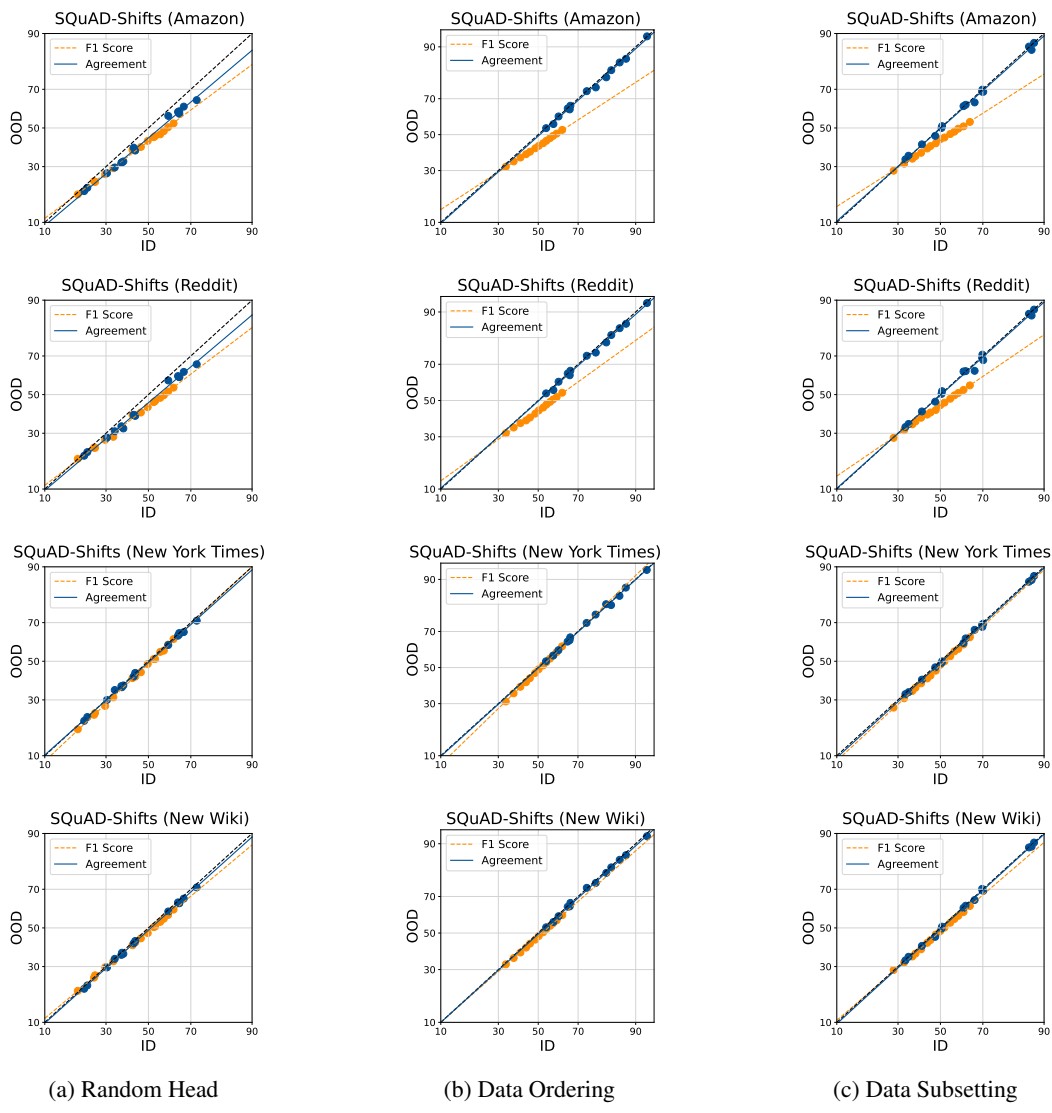

(a) Random Head        (b) Data Ordering        (c) Data Subsetting

Figure 29: ID vs OOD accuracy and agreement of models finetuned on SQuAD from a single pretrained OPT-125m with 10% of the training data

### A.4.4 Diversity in Full Finetuned BERT for Different Data Portions

We track the effect of diversity in random initialization, data ordering, and data subsetting for different portions of the training data ($100\%$, $50\%$, $30\%$, $10\%$). For each percentage $x\%$, Random Initialization and Data Ordering ensembles are trained on the same randomly sampled $x\%$ proportion of the data while each model in the Data Subsetting ensemble observe different randomly sampled $x\%$ portions.

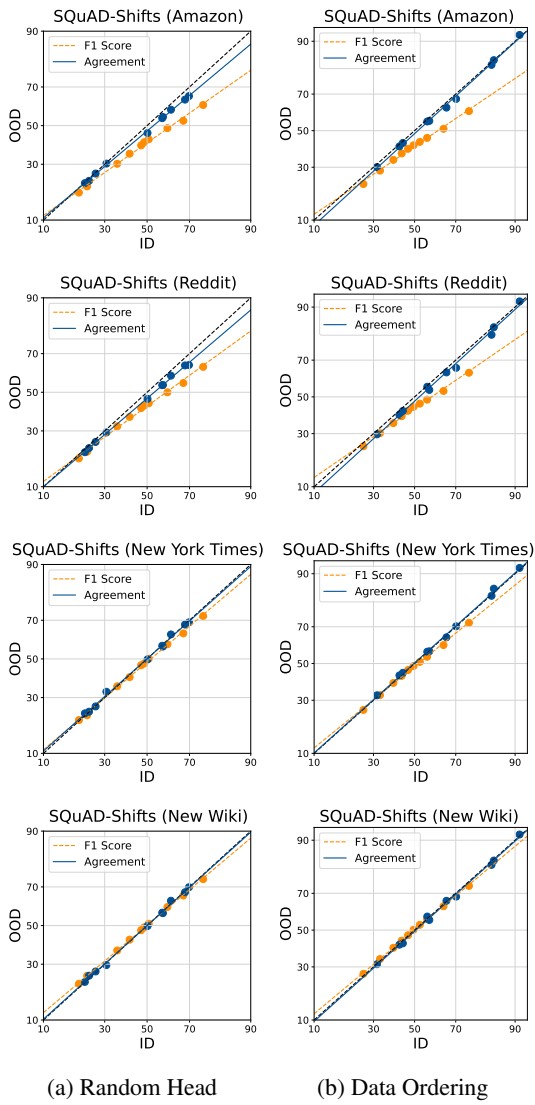

(a) Random Head   (b) Data Ordering

Figure 30: ID vs OOD accuracy and agreement of models finetuned on SQuAD from a single pretrained BERT with 100% of the training data

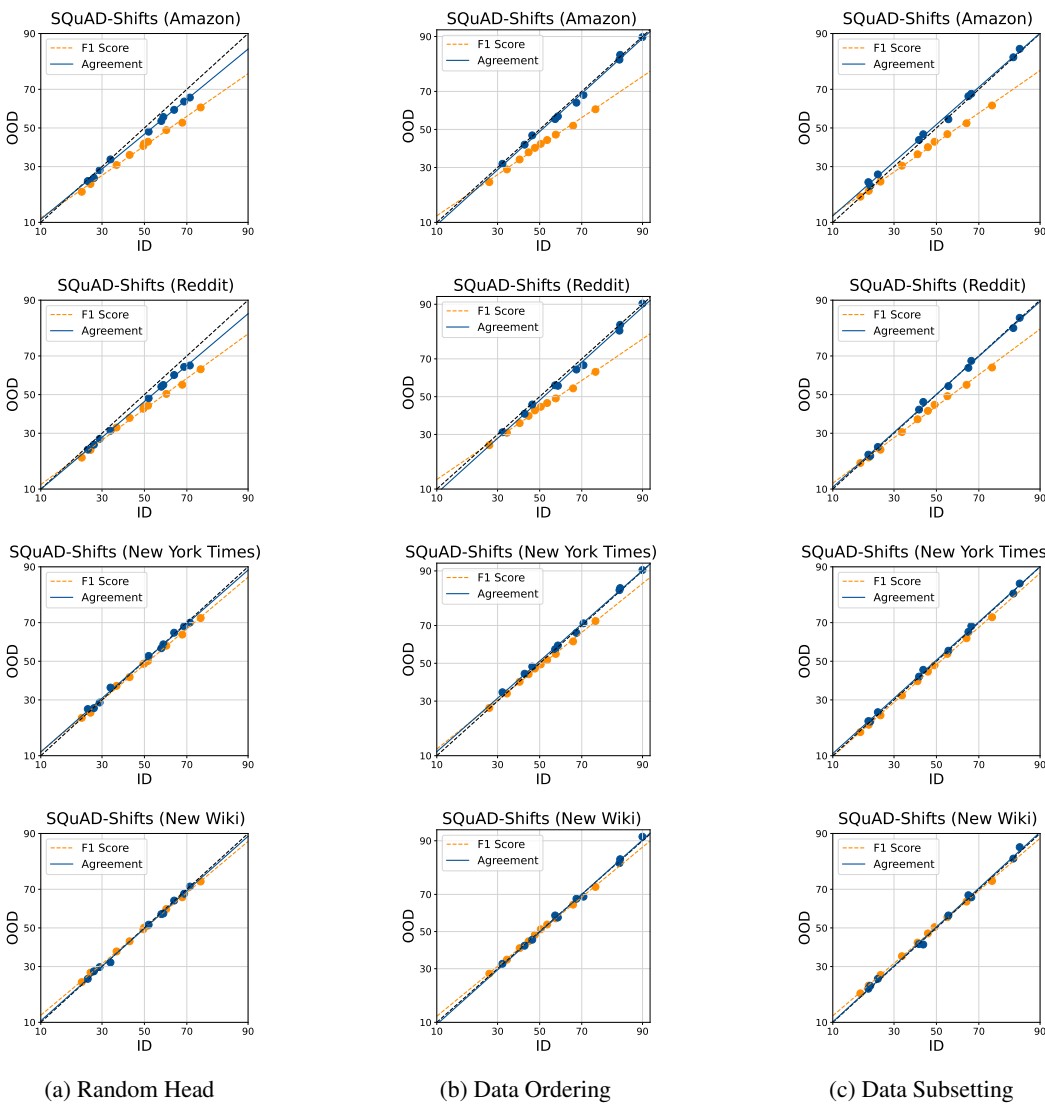

(a) Random Head        (b) Data Ordering        (c) Data Subsetting

Figure 31: ID vs OOD accuracy and agreement of models finetuned on SQuAD from a single pretrained BERT with 50% of the training data

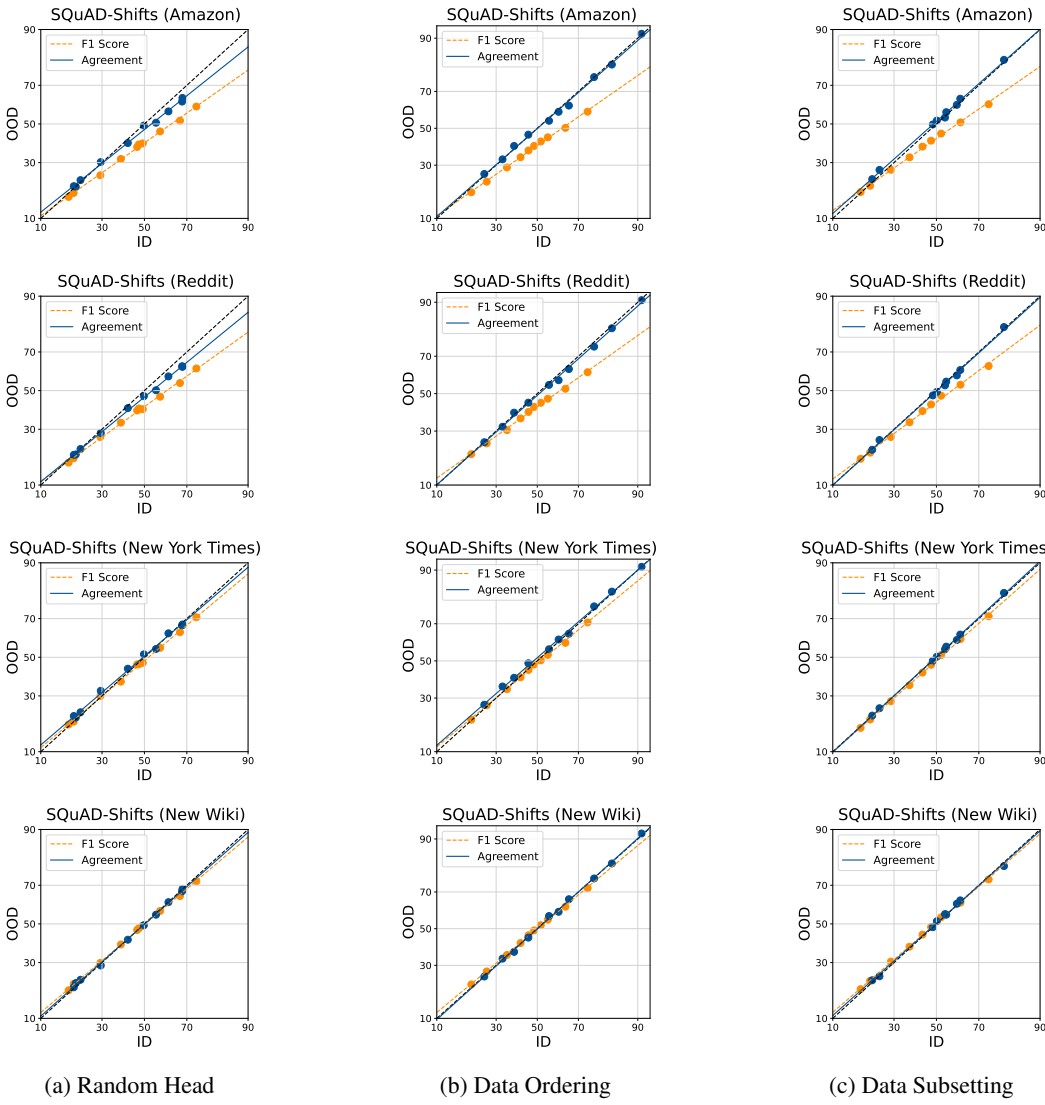

(a) Random Head      (b) Data Ordering      (c) Data Subsetting

Figure 32: ID vs OOD accuracy and agreement of models finetuned on SQuAD from a single pretrained BERT with 30% of the training data

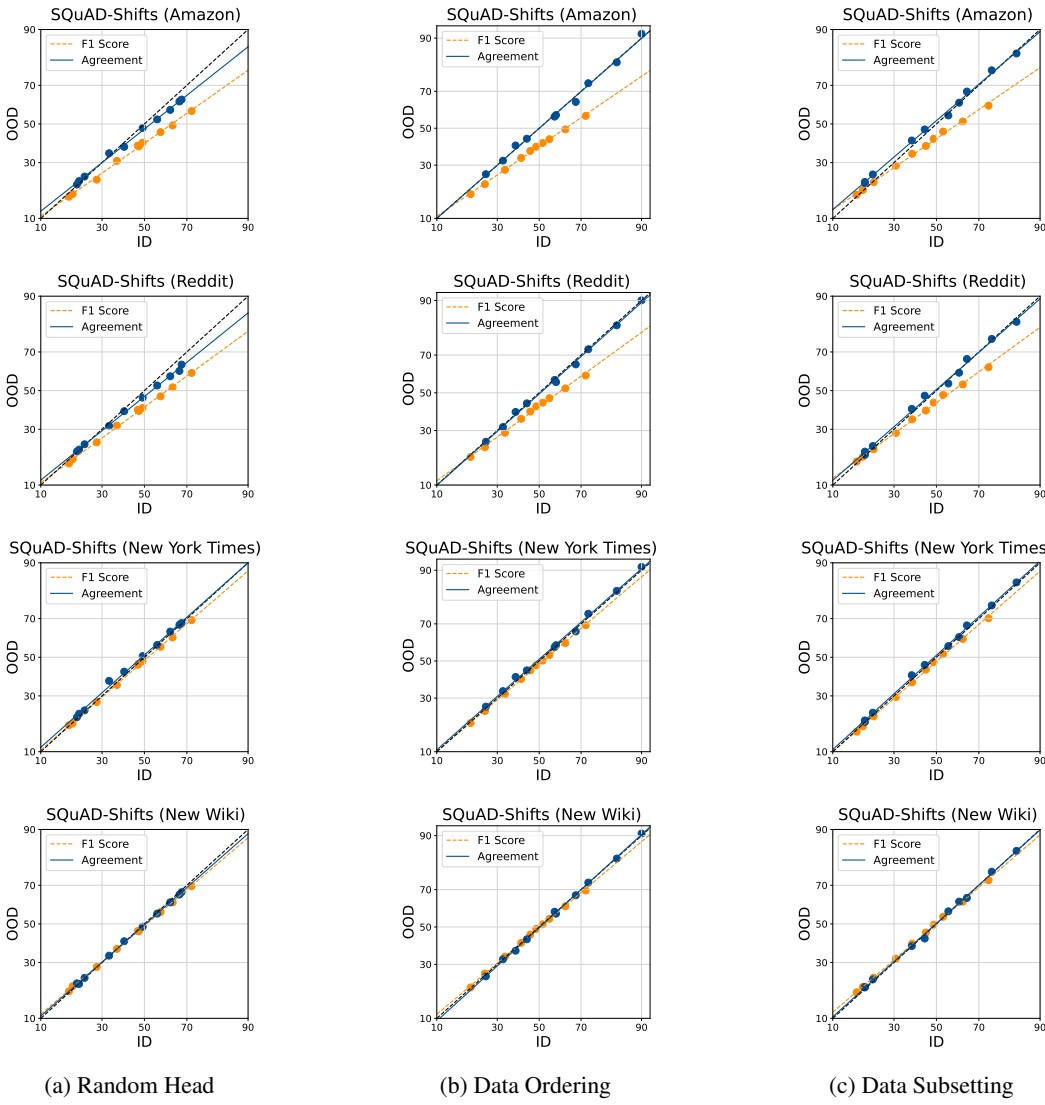

(a) Random Head  (b) Data Ordering  (c) Data Subsetting

Figure 33: ID vs OOD accuracy and agreement of models finetuned on SQuAD from a single pretrained BERT with 10% of the training data

### A.5 More Experiments on the Effect of Diversity Source for Text Classification

#### A.5.1 Diversity Source for Linear Probing

In Section 3.2, we reported the effect of diversity source for full finetuned OPT. Here, we demonstrate similar results on text classification shift from MNLI-matched to SNLI with linear probed GPT2-Medium (Figure 34) and OPT-125M (Figure 35). Similarly, random initialization observes the strongest AGL behavior. In Table 16, we report the average MAPE of OOD performance estimation using ALine across models.

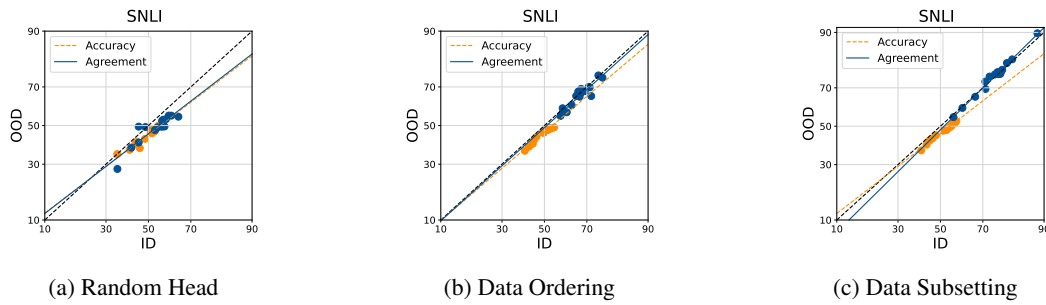

      (a) Random Head             (b) Data Ordering             (c) Data Subsetting

Figure 34: ID vs OOD accuracy and agreement of models finetuned on MNLI from GPT2-Medium. Random Head and Data Ordering ensembles are trained on $100\%$ of the training data while Data Subsetting ensemble is on $10\%$.

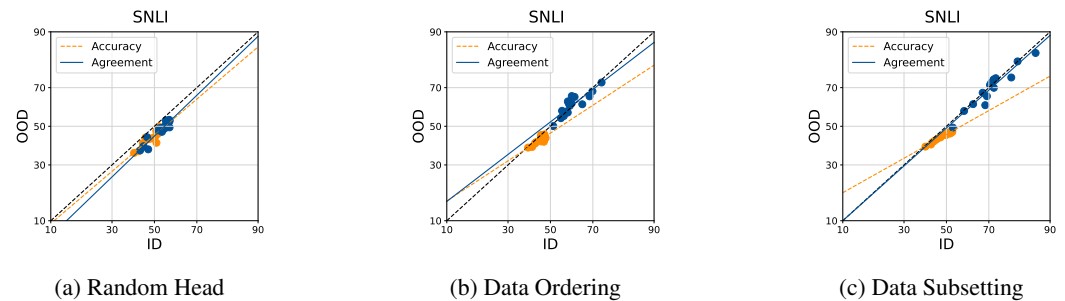

      (a) Random Head             (b) Data Ordering             (c) Data Subsetting

Figure 35: ID vs OOD accuracy and agreement of models finetuned on MNLI from OPT-125M. Random Head and Data Ordering ensembles are trained on $100\%$ of the training data while Data Subsetting ensemble is on $10\%$.

Table 16: The average MAPE (%) of ALine-D performance estimates of accuracy on SNLI.

| Model | GPT2-Medium | OPT-125M |
|---|---|---|
| Random Linear Heads | 5.9 | **5.4** |
| Data Ordering | 6.7 | 12.4 |
| Data Subsetting | **5.3** | 8.6 |

### A.6 Generative Question-Answering

We also test whether AGL appears for generative question answering, where models generate the answer to the question instead of directly extracting a span from the provided context.

#### A.6.1 Experiments on the effect of Diversity Source

For generative tasks, it is common to finetune over the base model directly instead of attaching a linear head on top. We again study different ways of achieving a diverse set of classifiers (e.g., Random Initialization, Data Ordering, Data Subsetting). However, we replace randomly initialization of a linear head instead with randomly initializing the non-zero initialized LoRA weights.

**Experimental Details**  We use a base model GPT2 and finetune models using cross-entropy loss on the next token prediction objective over SQuAD. During training, we concatenate the context, question, and answer together, and we apply the next token prediction objective on just the answer tokens. We add a line break ($\backslash n$) after the answer, which functions as a "stop token" that the models also have to predict after answering the question. We train with the AdamW optimizer with a learning rate of $1e-4$, weight decay of $1e-2$, and batch size of 16 up to a maximum of 4 epochs. As with extractive QA, we measure the F1 score of the model's answer (its output before a line break).

**Conclusion**  We track the effect of diversity for 100% of the training data in random initialization and data ordering, 50% of the training data in data subsetting. Figure 36 shows that all sources of diversity are sufficient to show AGL. Diversity is different from extractive question-answering because the search space of all tokens already provides enough diversity among models. We also note the MAE in Table 17 and MAPE in Table 18.

Table 17: ALine-S MAE (%) of different sources of diversity for generative question-answering on GPT2

| Source of Diversity | SQuAD-Shifts Amazon | SQuAD-Shifts Reddit | SQuAD-Shifts New-Wiki | SQuAD-Shifts NYT |
|---|---|---|---|---|
| Random Linear Heads | 0.0354 | 0.0170 | 0.0129 | 0.0148 |
| Data Ordering | 0.0296 | 0.0177 | 0.0165 | 0.0152 |
| Data Subsetting | 0.0312 | 0.0205 | 0.0178 | 0.0166 |

Table 18: ALine-S MAPE (%) of different sources of diversity for generative question-answering on GPT2

| Source of Diversity | SQuAD-Shifts Amazon | SQuAD-Shifts Reddit | SQuAD-Shifts New-Wiki | SQuAD-Shifts NYT |
|---|---|---|---|---|
| Random Linear Heads | 13.93 | 6.6 | 4.88 | 5.06 |
| Data Ordering | 12.05 | 6.57 | 5.91 | 5.41 |
| Data Subsetting | 16.43 | 10.08 | 7.47 | 8.32 |

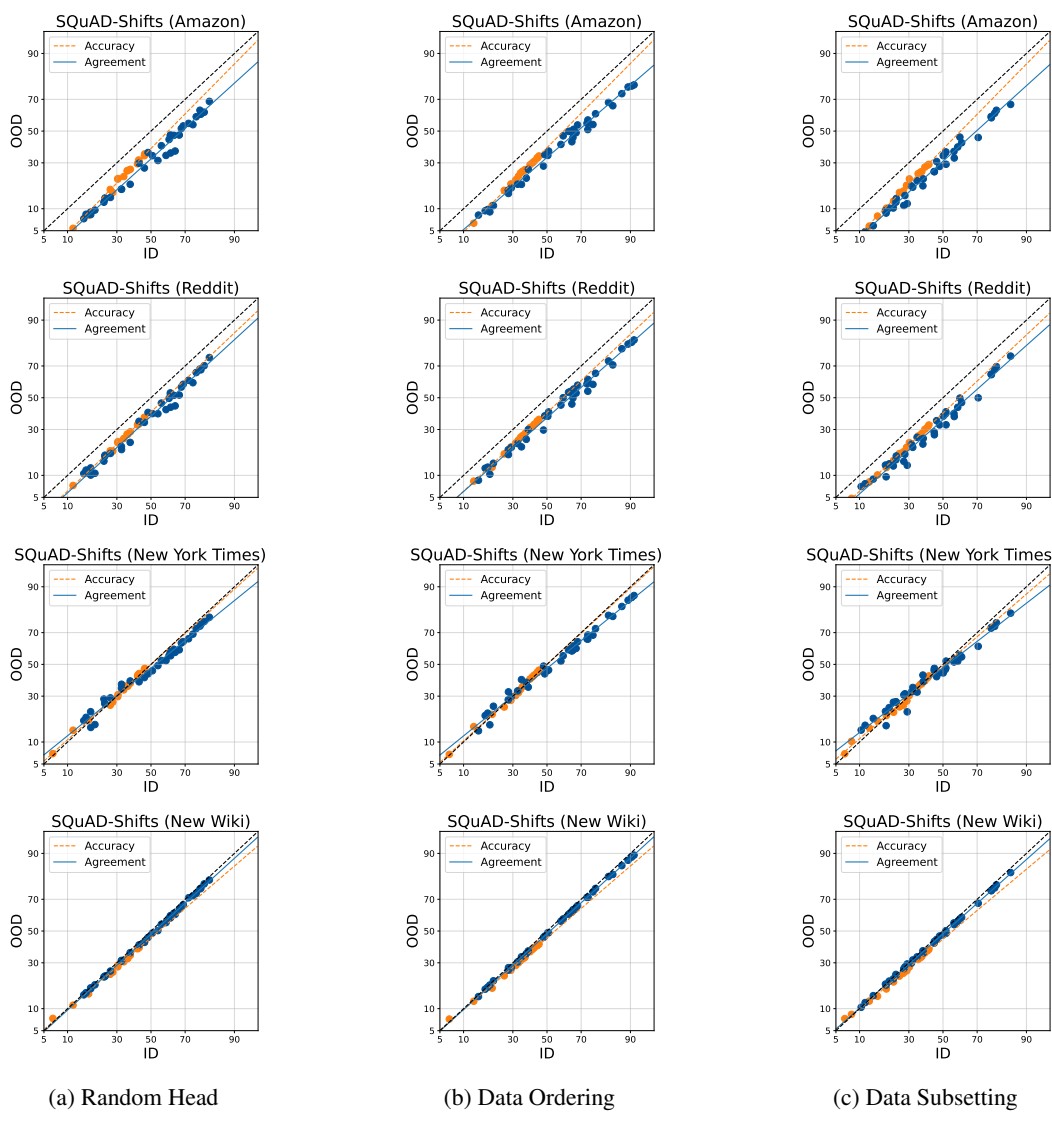

(a) Random Head       (b) Data Ordering       (c) Data Subsetting

Figure 36: ID vs OOD F1 and agreement of generative models finetuned on SQuAD from a single pretrained GPT2

### A.6.2 Experiments starting from multiple foundation models

We also finetune base models of the GPT and OPT model families for generative question-answering using the same hyperparameters mentioned in the previous section. Similarly, we evaluate models using the F1 score. Figure 37 shows that AGL holds for all shifts in SQuAD-Shifts. Furthermore, it also holds for the `mlqa-translate-test.es` test split of the MLQA dataset [32], which is the English portion of English-Spanish translated MLQA questions.

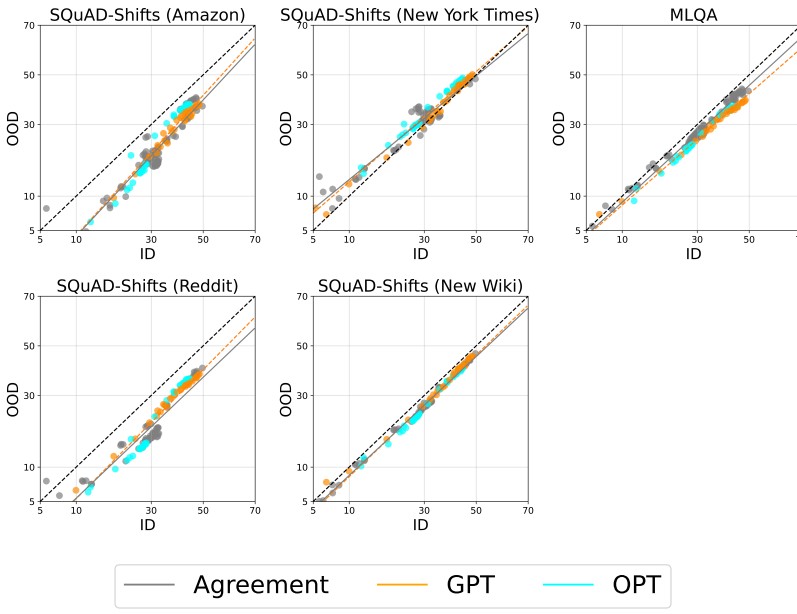

Figure 37: Generative models finetuned from different base models of the GPT and OPT family. AGL holds for all SQuAD-Shifts splits and MLQA Spanish split.

## A.7    Diversity under different learning rates and batch sizes

We test the robustness of our observation across learning rate and batch size for single base FM on CLIP embeddings. Figure 38 shows that AGL holds regardless of the learning rate or batch size used for fine-tuning. That is, AGL holds with random initialization of the head and does when for shuffling the data subset or randomizing the data ordering.

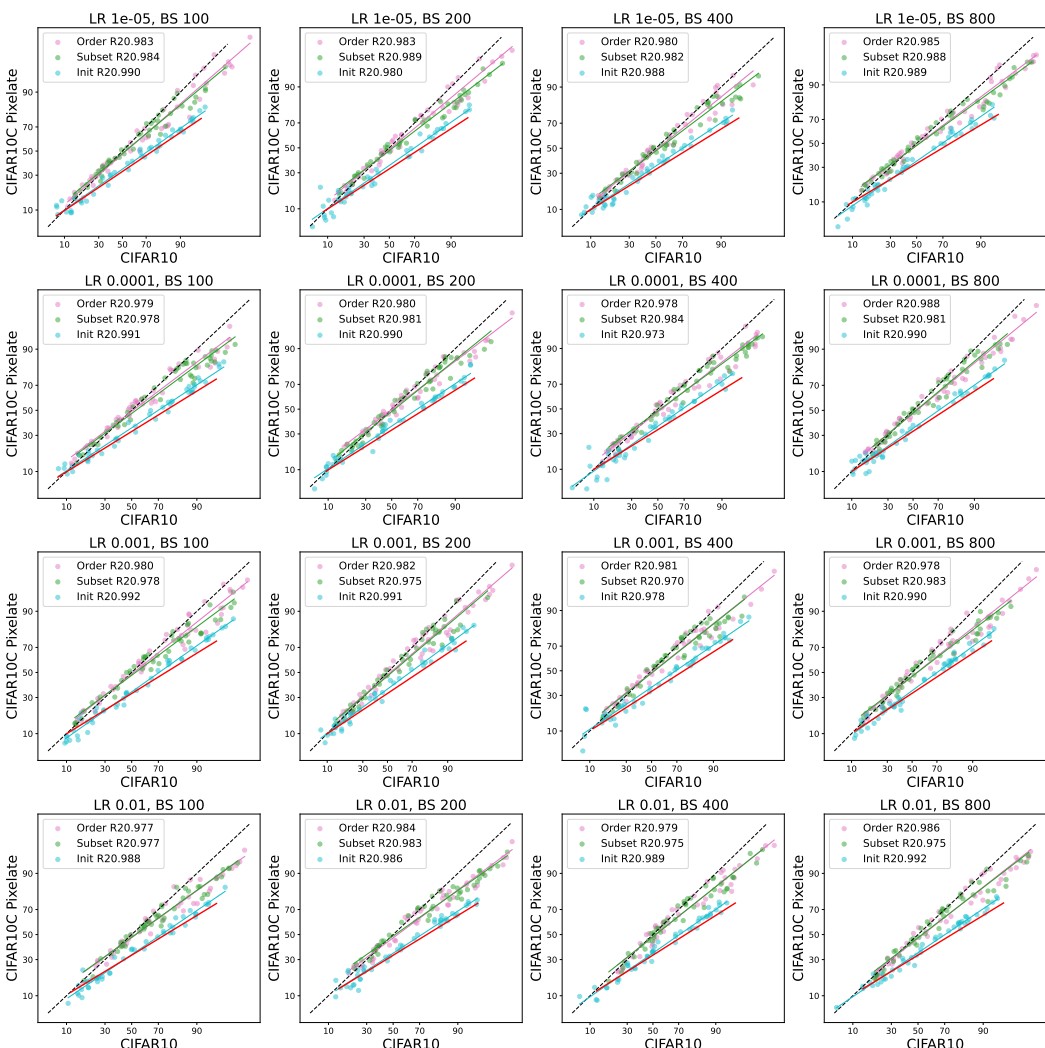

Figure 38: Over CLIP embeddings, we test the diversity sources under different learning rates and batch sizes. While all models lie on the same ID versus OOD accuracy line (red), only the set of models trained with different random initialization (cyan) achieves ID versus OOD agreement with the same slope and bias as ID versus OOD accuracy.

### A.8 Diversity Experiments for different PEFT methods

We compare full fine-tuning with different PEFT methods and observe that AGL holds for randomly initialized heads and does not for data ordering or data subsetting. Figure 39 shows a single GPT2 trained with LoRA [25], IA3 [33], and BitFit [64]. The linear fit for the three PEFT methods aligns with full fine-tuning for randomly initialized heads.

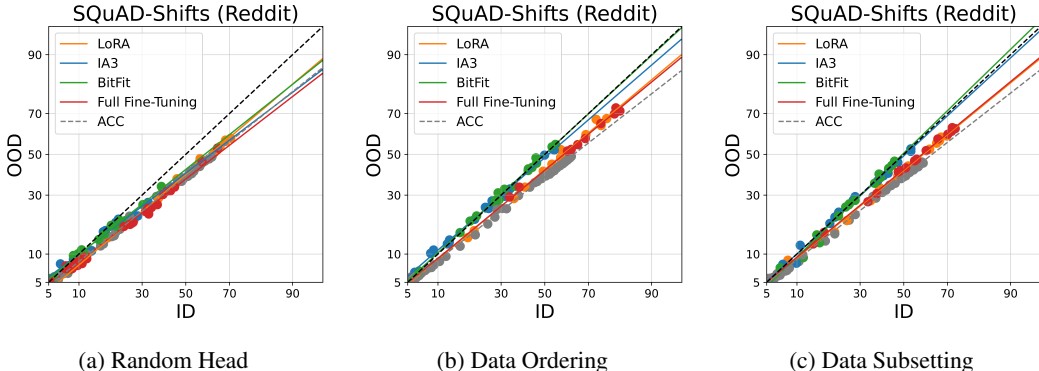

(a) Random Head          (b) Data Ordering          (c) Data Subsetting

Figure 39: ID (SQuAD) vs OOD (SQuAD-Shifts Reddit) Agreement trend from a single GPT2 for different PEFT methods (LoRA, IA3, BitFit). The accuracy line of all methods combined is shown in dotted gray while the agreement lines of each method is drawn as separate colors.

