# OpenReview forum: "Predicting the Performance of Foundation Models via Agreement-on-the-Line"
_NeurIPS.cc/2024/Conference — NeurIPS 2024 poster_

### Official Review · Reviewer_H4MK · 2024-06-22

**Soundness:** 2
**Presentation:** 3
**Contribution:** 2
**Rating:** 5
**Confidence:** 3

**Summary:**

The paper shows that agreement-on-the-line holds in finetuned foundation models across vision and language benchmarks and finds that random head initialization is critical for the phenomena. The authors also show that agreement-on-the-line holds in ensembles of different pretrained models. They demonstrate usage of agreement-on-the-line to predict OOD performance.

**Strengths:**

The paper is overall clear and well-written with substantial and thorough experiments on phenomenon of agreement-on-the-line across various language and vision benchmarks. The question of understanding and predicting model performance under distribution shifts is important in the field. The authors show that the proposed method of using agreement-on-the-line to predict OOD performance outperforms previous methods in most cases.

**Weaknesses:**

Minor points:
1.	The panel labels in Fig. 31 and 32 has a repetition of Random Head, which should be corrected.
2.	The authors fail to discuss the robustness of the method. Specifically, it is important to know how robust the method is in terms of changing different hyperparameters.

Major points:
1.	Novelty: the authors extend a previously proposed method (Baek et al. 2022) to finetuned foundation models. Although the aspect that observing AGL holds in this new setup is to some extent is novel, the methodology itself lacks novelty.
2.	The paper is mostly experimental results without providing any insights into why one would expect AGL to hold or correlate with ACL. Without any explanation or theoretical guarantee, it is debatable whether one can trust this method with new models or new data.

**Questions:**

1.	If you only train a linear head and freeze all the other parameters, isn’t the problem convex? Then, suppose training to convergence, initialization could potentially bias the final solution. However, this bias should be a result from the random initialization in the data kernel space. Why would this type of randomness be meaningful at all?
2.	What do you think are potential causes for the larger slopes of AGL compared to ACL if trained with data ordering or data subsetting?

**Limitations:**

The authors have listed limitations at the end. However, I think one major limitation is when the correlation coefficient is low, other methods will outperform AGL, which is shown in Table 5. I suggest authors include this limitation.

---

> ### Author Rebuttal · Authors · 2024-08-07
>
> >__The panel labels in Fig. 31 and 32 has a repetition of Random Head, which should be corrected.__
>
> Thank you for catching this! The third panel (c) corresponds with Data Subsetting, not Random Head. We will correct this in the next version.
>
> >__…it is important to know how robust the method is in terms of changing different hyperparameters.__
>
> Thank you for your feedback! None of the hyperparameters we use to linear probe or fully-finetune the models in this work have been picked to observe this phenomena. In Figure 3 in our rebuttal PDF, we see that across different learning rates and batch sizes, linear models over CLIP only observe AGL by varying the random initialization.
>
> >__Novelty: the authors extend a previously proposed method (Baek et al. 2022) to finetuned foundation models. Although the aspect that observing AGL holds in this new setup is to some extent is novel, the methodology itself lacks novelty.__
>
> The novelty of our work stems from several factors.
>
> *First, OOD performance estimation of FMs is a critical problem.* Foundation models are utilized for a variety of downstream tasks and to safely use them, it is important that we have ways to estimate their OOD performance, especially in limited label settings. This problem is largely underexplored and we still lack an understanding of when older methods transfer.
>
> *Second, before this work, it was unintuitive whether AGL holds in FMs.* The importance of diversity source in finetuned FMs, in contrast to deep ensembles trained from scratch, is quite surprising. This has interesting theoretical implications.
>
>  1. Finetuning on top of pretrained weights can produce models with surprising levels of decorrelated errors when trained from different randomly initialized heads.
>
>  2. AGL can hold in linear probes, in contrast to Baek et al. which argue AGL only occurs in neural networks.
>
> We believe these findings would be interesting to both practitioners and theoreticians.
>
> *Third, our method can effectively estimate OOD performance for a wider range of scenarios than other baselines, which break for non-classification tasks.*  In QA benchmarks, the prediction error is 20% smaller than popular confidence-based prediction methods.
>
> >__The paper is mostly experimental results…Without any explanation or theoretical guarantee, it is debatable whether one can trust this method with new models or new data.__
>
> Thank you for the feedback! While we do not provide exact theoretical guarantees, the conclusions we make about ensemble diversity in FMs and their effect on observing AGL/ACL hold across hundreds of finetuned models we tested from different model familes (GPT, OPT, Llama), 50+ distribution benchmarks (Appendix A.3 and A.4), hyperparameters (see learning rate/batch size sweep in rebuttal Figure 3), finetuning strategies (LP, LoRA, full finetuning; other PEFT methods in rebuttal Figure 1). We hope that our rigorous experimental report can demonstrate that ACL/AGL is a powerful tool for predicting the performance of FMs.
>
> Furthermore, the coefficient of determination $R^2$ of ID vs OOD agreement (see Sec. 5) gives practitioners a rough guarantee for when the performance estimate derived by AGL is reliable. This is immensely useful in comparison to other confidence-based baseline methods which also do not come with any formal guarantees.
>
> We’d also like to point to several theoretical works that have tried to characterize when ACL and AGL hold [1, 2]. We believe our study on the importance of random initialization for observing AGL introduces a new conceptual angle for understanding deep ensembles and what actually induces AGL. We hope this can inspire further theoretical research on this important topic.
>
> [1] Mania and Sra. Why do classifier accuracies show linear trends under distribution shift? 2020.
>
> [2] Lee, et al. Demystifying Disagreement-on-the-Line in High Dimensions. 2023.
>
> >__If you only train a linear head, isn’t the problem convex? Then, suppose training to convergence,…(bias) result from the random initialization in the data kernel space. Why would this type of randomness be meaningful?__
>
> >__What are potential causes for the larger slopes of AGL compared to ACL if trained with data ordering or data subsetting?__
>
> Thank you for the great question! In all of our ID versus OOD scatter plots, you will see that we have models of a wide range of accuracies. This is because in addition to the diversity source, these models are also _finetuned for different timesteps_. In fact, a lot of these “partially” trained models can be close to their initialization, and any bias from random initialization isn’t restricted to the kernel of the training data. However, the bias in the kernel induced by random initialization may be important for achieving the low _OOD_ agreement rates. We provide some intuition below.
>
> Consider the linear probe setting wherein we train a linear classifier on top of fixed features. Suppose we assume that the ID training data is low rank. Let S be the span of the ID training data, and N be the corresponding null-space that has some overlap with the OOD data. The training iterations only update the classifier in S. As a result, under different random initializations, classifiers would retain their different initializations on S and lead to lower OOD agreement. However, if the initializations are fixed, and only the data ordering or subset varies, these orthogonal components no longer vary, leading to higher OOD agreement. Intuitively, such an argument should carry over to heavy-tailed training data as well, beyond low-rank. We were unable to come up with an analogous interpretable diversity introduced by data-subsetting or ordering.
>
> Overall, the phenomenon of agreement-on-the-line is poorly understood, and we think our work reveals new empirical observations that help form a better picture. We believe our work would inspire and inform future work that rigorously and comprehensively explains AGL.

---

> > ### Comment · Reviewer_H4MK · 2024-08-09
> >
> > I thank the authors for their detailed responses! The authors have addressed my concerns around robustness with respect to the hyperparameters. Thus, I am increasing my score from 4 to 5. However, I still hold my skeptical position. Without any explanations or theoretical insights, I am doubtful whether AGL (particularly given its origin from random initialization in a convex problem) is indeed meaningful or just a mere coincidence that has trivial explanation.

---

### Official Review · Reviewer_8Ly8 · 2024-06-22

**Soundness:** 3
**Presentation:** 3
**Contribution:** 3
**Rating:** 7
**Confidence:** 3

**Summary:**

The paper studies the applicability of agreement-on-the-line (AGL) to finetuned large models. Specifically, AGL is the phenomenon where the agreement in predictions of a collection of models on in-distribution (ID) data is linearly correlated with these models' agreement on out-of-distribution (OOD) data. This phenomenon is particularly interesting since earlier work has shown that this relation holds whenever the accuracy-on-the-line (ACL) holds; furthermore, these linear relation for agreement and accuracy is the same. While prior work has studied this phenomenon in a variety of settings, this paper studies instead this phenomenon for large finetuned models. The authors show that even in this regime, AGL holds in both vision and language settings. Furthermore, while prior work has shown that vision models pretrained on different distributions do not share the same AGL line, the authors find that the agreement of language models actually falls on the line.

**Strengths:**

- The paper is well-written.
- Generalizing the AGL phenomenon to finetuned large models is important as the use of these models becomes prevalent.
- The authors run extensive evaluation on models of different sizes and on different downstream tasks.

**Weaknesses:**

- The paper lacks an analysis on the reason behind the different behavior observed between the vision and language models. Although these are two different modalities, I don't see a reason why vision and language models should behave differently.
- The paper lacks an analysis of AGL in zero/few-shot settings.
- The paper assumes that full finetuning and finetuning with LoRA lead to the same AGL phenomenon. While LoRA (and other PEFT methods) lead to a similar performance, their behavior might be different [1]. Such an analysis is needed, and the inclusion of more PEFT methods (QLoRA, BitFit, IA3, etc.) would be nice.


[1] Empirical Analysis of the Strengths and Weaknesses of PEFT Techniques for LLMs. Pu et al. 2023.

**Questions:**

- The authors point out that they vary several factors when finetuning the large models, including the initialization of the linear head. I am a bit confused why different initialization would lead to substantially different accuracies. For example, in figures 1 and 2 (where there's supposed to be a single backbone that's finetuned several times), we can see that the x-axis values range from 10% to 90%. Can you please explain the reason behind this wide range?
- Can you please add more results with different PEFT methods and analyze the difference (or the similarity) between them?

**Limitations:**

Yes

---

> ### Author Rebuttal · Authors · 2024-08-07
>
> Thank you for your valuable comments and suggestions! We address each of your concerns below in detail.
>
> >__The paper lacks an analysis on the reason behind the different behavior observed between the vision and language models.__
>
> The conclusions we make in this work apply to _both_ image and language models, we do not differentiate between these settings in any way. Models of both modalities require special attention to the source of diversity when fine-tuning from a common pretrained checkpoint to observe AGL. In particular, the diversity induced via random head initialization yields AGL, while the diversity induced via data reordering or data subsetting does not result in AGL.
>
> The only differences we do make between vision and language tasks are
>
> __1)__ the fine-tuning strategy that we employ - e.g. full fine-tuning for question answering (since linear probing performs very poorly), and linear probing for image classification
>
> __2)__ the choice of evaluation metric (F1 score for QA and 0-1 accuracy for classification).
>
> As emphasized in Section 2, our diversity results hold across tasks, regardless of what fine-tuning strategy or metric is employed. We will make this more clear in the next version.
>
> >__The paper lacks an analysis of AGL in zero/few-shot settings.__
>
> Thanks for the feedback! We have some preliminary experiments on few-shot and zero-shot learning to answer your question in Figure 2 of our rebuttal PDF. We will add these results to the paper if you find that it strengthens our work.
>
> We consider few-shot linear probing over CLIP features, where we train on 10 examples in CIFAR10 per class and test models on the shift CIFAR10C Pixelate. Similar to fine-tuned FMs, we see that _by varying the random initialization of the linear probes, we can observe AGL and ACL._ On the other hand, data subsetting and data reordering trivially do not work in this setting as there’s only a small handful of training examples.
>
> The zero-shot setting is a vastly different regime for understanding ACL and AGL because any downstream task is “out-of-distribution”. In future work, it may still be interesting to study the linear relationship between the pretraining loss (ID) versus the downstream task loss (OOD), or simply between two OOD tasks e.g., downstream task 1 versus downstream task 2. In Figure 2 of the rebuttal PDF, we take the intermediate pretraining checkpoints of OLMo 7B and evaluate their zero-shot performance on SQuAD versus SQuAD-shifts Reddit. On the contrary to finetuned models, we see that both ACL and AGL do not hold in this setting.
>
> >__The paper assumes that full finetuning and finetuning with LoRA lead to the same AGL phenomenon. While LoRA (and other PEFT methods) lead to a similar performance, their behavior might be different [1]. Such an analysis is needed, and the inclusion of more PEFT methods (QLoRA, BitFit, IA3, etc.) would be nice.__
>
> >__Can you please add more results with different PEFT methods and analyze the difference (or the similarity) between them?__
>
> Thank you for the suggestion! We first want to clarify that we do not claim that PEFT and full-finetuned models behave the same generally or that they always lie on the same ACL and AGL trend. For example, other works have reported circumstances where PEFT and full-finetuned models observe different levels of effective robustness under distribution shift (i.e., different ACL slopes) [1]. The reason we group these two finetuning methods together in our work is mostly for notational convenience (i.e., LP versus FFT) since across the datasets we evaluate in our work, PEFT and full-finetuned models do observe the same effective robustness. We will make this more clear in Section 2.
>
> To further strengthen our work, we have included additional experiments that directly compare different PEFT methods. We trained GPT2 for random head initialization, data ordering, and data subsetting with LoRA, IA3, and BitFit as shown in Figure 1 of the rebuttal PDF. Regardless of the PEFT method, the accuracy points lie on the same line and AGL holds best with random head initialization. Interestingly, this indicates that even with different PEFT methods, we can observe AGL in ensembles as long as the linear head is randomly initialized.
>
> [1] Chen, et al. Benchmarking Robustness of Adaptation Methods on Pre-trained Vision-Language Models. Neurips 2023.
>
>
> >__I am a bit confused why different initialization would lead to substantially different accuracies. For example, in figures 1 and 2…we can see that the x-axis values range from 10% to 90%. Can you please explain the reason behind this wide range?__
>
> Good question! The large range of accuracies is a consequence of models being trained for varying amounts of training epochs and this is necessary to observe the full ACL/AGL linear trends. For each source of diversity we also vary the number of epochs (not train until convergence) to obtain this range of accuracies. As we finetune from a randomly initialized linear head, the model performance at the beginning of finetuning is almost random (10% for CIFAR10 classification).

---

> > ### Comment · Reviewer_8Ly8 · 2024-08-09
> >
> > Thank you for the clarification and for running additional experiments. Adding these results to the main paper (or in the Appendix and referencing them in the main paper) would definitely strengthen the paper. I will raise my score to 7.

---

### Official Review · Reviewer_4CKp · 2024-07-09

**Soundness:** 3
**Presentation:** 3
**Contribution:** 2
**Rating:** 7
**Confidence:** 3

**Summary:**

The authors of the paper propose a method to demonstrate that foundation models can exhibit agreement-on-the-line (AGL), under certain conditions. The existence of AGL can be used to predict the OOD capabilities of models without having access to the labels for the downstream tasks.

**Strengths:**

- While AGL has been observed in the literature, the paper is novel (to the best of my knowledge) in applying it to the setting of foundation models.

- The subject of the paper is also interesting - measuring the OOD performance of models is an important topic, and this paper does so with the added benefit of not explicitly requiring downstream labels (which, as the authors note, may be difficult to procure).

- The experiments done by the authors are convincing, for the most part. The situations where foundation models exhibit AGL are clear, and it is easy to understand how the existence of AGL translates to good predictions about OOD performance.

**Weaknesses:**

- I believe that the paper's clarity, while overall good, could be improved further in certain points:

  - Figure 1 should be a little clearer, with the caption being a little bit more detailed. This will help a lot with understanding of the key results of the paper, given the early position of the Figure in the document.

  - I feel like the authors should elaborate a bit on lines 246 - 255, given that their result here is in contrast with previous statements made in the literature. A little more discussion here would be helpful.

  - Similarly, Figure 3 should also be expanded a bit more, especially since in some settings the ID - OOD line lies directly on top of the $y= x$ axis, which I find very surprising.

- The fact that AGL can predict OOD performance, while interesting, comes with the caveat in Section 5 that it cannot be currently applied to all datasets. The authors explicitly state that and provide a criterion to determine the setting in which AGL is predictive of OOD accuracy. Nevertheless, I think this Section requires a bit more detail (see also my question below).

Overall, this is an interesting paper in my opinion, and I think my concerns with it currently are mostly based on the clarity.

**Questions:**

I would like the authors to explain what the correlation $R$ in Section 5 refers to. If it is ID vs OOD agreement for various models, then it would mean that these should be nearly parallel to the $y = x$ line, which is a limitation of the setting where AGL is predictive of OOD performance.

**Limitations:**

I think the authors have adequately addressed the limitations of their work, and I cannot find any negative societal impact arising from their work.

---

> ### Author Rebuttal · Authors · 2024-08-07
>
> Thank you for your valuable comments and suggestions! We address each of your concerns below in detail.
>
> >__Figure 1 should be a little clearer, with the caption being a little bit more detailed. This will help a lot with understanding of the key results of the paper…__
>
> Thank you for the suggestion! We will update the Figure 1 caption if accepted to:  “The ID vs OOD lines for accuracy (yellow) and agreement (blue) for various datasets and fine-tuned ensembles. Each blue dot corresponds to a member of the ensemble and represents its ID (x) and OOD (y) accuracy; and each yellow dot corresponds to a pair of these members and represents their ID (x) and OOD (y) agreement. From CIFAR10 to CIFAR10C “Pixelate”' in linear probed CLIP, MNLI to SNLI in full fine-tuned OPT, and SQuAD to SQuAD-Shifts “Amazon” in full fine-tuned GPT2, we observe different agreement linear fits depending on the diversity source (columns) used to generate the ensemble.”
>
> >__I feel like the authors should elaborate a bit on lines 246 - 255, given that their result here is in contrast with previous statements made in the literature.__
>
> Thank you for the suggestion! In our work, we demonstrated that _random initialization_ in particular is important to observe AGL in fine-tuned FM ensembles. We tie this finding to previous literature in lines 246-255 that study how well different sources of diversity in deep ensembles induce a related but different phenomena called GDE.
>
> In particular, GDE is a phenomenon in deep ensembles (neural networks, random forests) where ID accuracy tends to equal ID agreement exactly [1, 2]. Works have shown that, beyond the classical ensembling technique of bagging / data subsetting, deep ensembles induced by varying the data or model seed (ordering / initialization) can also induce this equality.
>
> On the other hand, in our problem setting, we found that ensembles induced by different random initialization achieves AGL, while data ordering / subsetting cannot. Our setting is different from previous literature in two distinct ways:
>
> __1.__ AGL studies the _OOD_ agreement rate relative to their ID agreement, in contrast to the GDE phenomena which only regards the models’ ID agreement. We hypothesize that random initialization is much more important for observing the right levels of OOD agreement.
>
> __2.__ Models are only _lightly fine-tuned_ or linearly probed, unlike deep ensembles trained from scratch. Diversity sources may behave differently in this circumstance.
>
> We will add this further discussion in the camera ready.
>
> [1] Jiang, et al. Assessing Generalization of SGD via Disagreement. ICLR 2022.
>
> [2] Nakkiran and Bansal. Distributional Generalization: A New Kind of Generalization. Preprint 2020.
>
> >__Figure 3 should also be expanded a bit more, especially since in some settings the ID - OOD line lies directly on top of the y=x axis, which I find very surprising.__
>
> Thank you  for your suggestion! We will make sure to expand on the main takeaways from Fig 3. Notably, we observe that the agreement rate between models from different model families (e.g., between GPT and Llama models) observe agreement-on-the-line across different NLP tasks.
>
> Indeed, there are certain shifts such as from SQuAD to SQuAD-shifts New Wiki and SQuAD-shifts NYT where model performance barely drops. This is partially due to the large-scale pretraining, but also the distribution shift is simply much smaller – SQuAD was constructed using Wikipedia, so looks closer to SQuAD-Shifts New Wiki and NYT than Amazon reviews and Reddit [1].
>
> What is most peculiar is that ID vs OOD agreement tracks the linear trend of ID vs OOD accuracy accordingly. For small distribution shifts SQuAD-Shifts New Wiki and NYT, the models’ agreement ID vs OOD is also close to being y=x. For larger shifts such as SQuAD-shifts Amazon and Reddit, ID vs OOD agreement also moves away from the y=x line.
>
> We will make these clarifications in Figure 3.
>
> [1] The Effect of Natural Distribution Shifts on Question Answering Models. Miller et al. 2020.
>
> >__I would like the authors to explain what the correlation R in Section 5 refers to. If it is ID vs OOD agreement for various models, then it would mean that these should be nearly parallel to the  y=x line…__
>
> Thank you for the feedback! The $R^2$ we mention in Section 5 refers to the _standard linear regression coefficient of determination_. $R^2$ ranges between 0 and 1 and it measures how well the relationship between two variables, $X$ vs $Y$, can be explained by a linear function. When $R^2$ is high, $Y$ can be estimated as $aX + b$ with small residual error. In our paper, we use $R^2$ to measure the strength of the linear correlation in ID vs OOD agreement and ID vs OOD accuracy.  Note that $R^2$ is different from the slope of the linear fit. For example, consider SQuAD-shifts Reddit in Figure 3, both ID vs OOD accuracy and agreement have high $R^2$ values, but the slope is far away from the $y=x$ line.
>
> According to Baek et al. [1], $R^2$ can determine when AGL holds. In particular, when ID vs OOD agreement strictly follows a linear trend (i.e., the linear correlation has high $R^2 > 0.95$), then ACL also holds with the _same slope and bias_. In such circumstances, we can use AGL-based methods ALineS and ALineD to estimate the OOD performance of models precisely without labels. Interestingly, as we have demonstrated in our paper, many natural distribution shifts across image/text classification and QA observe AGL and ACL with high $R^2$.
>
> We will make these points clearer in Section 5 of the final draft.
>
> [1] Baek et al. Agreement-on-the-line: Predicting the performance of neural networks under distribution shift. NeurIPS 2022.

---

> > ### Comment · Reviewer_4CKp · 2024-08-12
> > **Response to rebuttal**
> >
> > Thank you very much for the detailed response to my comments! As all of the points I made have been addressed, I am raising my score a bit.

---

### Official Review · Reviewer_hhqs · 2024-07-12

**Soundness:** 3
**Presentation:** 3
**Contribution:** 3
**Rating:** 7
**Confidence:** 3

**Summary:**

This submission studies the problem of predicting OOD performance given known in-domain performance. Building on top of recent work showing that ensembles can be used for this problem, by looking at agreements between components in the ensemble as surrogate labels to predict OOD performance, they find that a similar approach can be used for finetuned LLMs, as long as ensembles are generated by finetuning randomly initialized heads. The topic is timely, the paper is generally clearly written (although improvements are needed) and the empirical validation is good.

**Strengths:**

+ relevant and important topic
+ simple and practical approach

**Weaknesses:**

- some critical details are missing, for instance the authors should report in the main paper how to go from Acc, Agr to OOD (line 160-164).
- the significance is unclear to me: It would be useful to predict OOD performance at larger scales as opposed for models that are finetuned for more steps.

**Questions:**

If only the topmost layer is randomly initialized and trained, then all components of the ensemble should converge to the same parameter vector provided that they are trained for long enough because the optimization is a convex problem. Is the diversity due to the limited number of training step? If it is the case, I think the assumption should be made explicit.

Could the authors compare the method of sec. 4 against the method of sec 3 directly?

**Limitations:**

no concern

---

> ### Author Rebuttal · Authors · 2024-08-07
>
> Thank you for your valuable comments and suggestions! We address each of your concerns below in detail.
>
> >__Some critical details are missing, for instance the authors should report in the main paper how to go from Acc, Agr to OOD (line 160-164).__
>
> We apologize for the lack of clarity. The default prediction algorithm (ALineS) used to estimate OOD accuracy when agreement-on-the-line holds, is as follows:
>
> 1. Say there is a linear trend in ID versus OOD agreement, i.e., for any two pairs of models $w, w’$ we have
> $\Phi(Agr_{OOD}(w, w’)) = a * \Phi(Agr_{ID}(w, w’)) + b$
>  where $\Phi(\cdot)$ is the probit scaling.
>
> 2. Estimate the slope and bias ($\hat{a}, \hat{b}$) of the above linear trend using OLS.
>
> 3. Apply this linear transformation to accuracy:
> $\Phi(Acc_{OOD}(w)) = a * \Phi(Acc_{ID}(w)) + b$
>
> We will update Section 2.3 to include equations defining accuracy-on-the-line and agreement-on-the-line in the main body. In addition, we have included a detailed discussion of the prediction algorithms (ALineS and ALineD) in Appendix A.1.1.
>
> >__The significance is unclear to me: It would be useful to predict OOD performance at larger scales as opposed for models that are finetuned for more steps.__
>
> Thank you for this important question! You can utilize our results in the multiple base model section (Section 4) as a type of “scaling law”. In Figure 3, we showed that finetuned models of a large range of model scales (125M to 7B parameters) and model families (GPT, Llama2, etc.) observe the same ACL and AGL trends in QA and text classification. Using smaller models such as GPT2, we can compute the ACL/AGL trend, and use this to project the OOD performance of larger models _without any labels_ given their ID performance.
>
> >__If only the topmost layer is randomly initialized and trained, then all components of the ensemble should converge to the same parameter vector…because the optimization is a convex problem. Is the diversity due to the limited number of training step?__
>
> As mentioned in Section 3, to construct the ensemble, we vary the number of epochs/training steps to get models with a wide range of ID accuracies. We will make this more clear in the revision - thank you for pointing this out!
>
> >__Could the authors compare the method of sec. 4 against the method of sec 3 directly?__
>
> Thank you for this question! To reiterate, Section 3 studies AGL in ensembles of models finetuned from a _single_ base foundation model, where we establish that randomly initializing the head is important for observing AGL. In Section 4, we show that ensembles of models fine-tuned from different base foundation models (i.e. LLama, GPT, OPT) also exhibit AGL.
>
> To answer your question, we do a direct comparison of AGL under these two scenarios. Specifically, we have a set of 14 finetuned GPT2 models, and we measure their agreement rate with
>
> *Setting 1:* other GPT2 models finetuned with different random initialization
>
> *Setting 2:* models finetuned from other base models (Llama, OPT).
>
> Using these agreement rates, we predict the OOD performances of the GPT2 models using the ALine-S algorithm. We report the MAE and MAPE for the two settings below. Both are quite effective with very small MAE’s (estimates OOD accuracy with error less than 2%).
>
> __SQuAD vs SQuAD-Shifts Reddit__
> |          | Setting 1 | Setting 2|
> |--------------- |--------------|--------------|
> |MAE| 1.27| 1.54|
> |MAPE| 5.19| 4.82|

---

> > ### Comment · Reviewer_hhqs · 2024-08-10
> > **thank you**
> >
> > I'd like to thank the authors for their response which I find satisfactory. I hope they will revise the paper accordingly. I am still supportive of accepting this paper.

---

### Author Rebuttal · Authors · 2024-08-07

We thank all reviewers for their great feedback and questions about our paper! The reviewers generally found our paper interesting and praised our work for proposing a simple yet effective solution to predict the OOD performance of FMs. Here, we briefly summarize the common concerns, and new experiments we’ve added in our rebuttal PDF:

__*How do you achieve a wide range of accuracies and agreements in the linear probing setting where the loss landscape is convex?*__ In each ensemble, we also vary the number of epochs we finetune each model. Regardless, the random initialization also has to vary to observe AGL.

__*Can we use this method to forecast OOD performance with larger scales?*__ Yes! We showed in Section 5 that on many language tasks, foundation models from different families (GPT, OPT, LLama) of different sizes all lie on the same ACL and AGL trends. This means we can estimate the linear trend using smaller models to extrapolate the performance of larger models with no labels OOD.

__*Is it interesting to study fine-tuned FMs?*__  It is common practice to finetune FMs by linear probing or LoRA, and we believe our study can apply to many practical use cases for OOD estimation. As requested, we also extend our study to few-shot and zero-shot settings in rebuttal Figure 2. In the few-shot setting, we also observe that ACL/AGL hold in CLIP by varying the random initialization. On the other hand, zero-shot language models do not necessarily observe AGL/ACL trends as strongly as fine-tuned models on SQuAD versus SQuAD-Shifts. Zero-shot models may behave differently as neither SQuAD nor SQuAD-Shifts is “in-distribution”.

__*Are there any theoretical guarantees we provide about ACL or AGL?*__ While we do not provide exact theoretical guarantees, the conclusions we make about ensemble diversity in fine-tuned FMs and their effect on observing AGL/ACL hold across hundreds of fine-tuned models we tested from different model families (GPT, OPT, Llama), 50+ distribution benchmarks (Appendix A.3 and A.4), hyperparameters (see learning rate/batch size sweep in rebuttal Figure 3), fine-tuning strategies (LP, LoRA, full fine-tuning; other PEFT methods in rebuttal Figure 1). We hope that our rigorous experimental report can demonstrate that ACL/AGL is a powerful tool for predicting the performance of FMs.

---

### Decision · Program_Chairs · 2024-09-25

**Decision:**

Accept (poster)

**Comment:**

This paper investigates the applicability of agreement-on-the-line (AGL) to finetuned large language models for predicting out-of-distribution (OOD) performance. Reviewers generally found the paper to be well-written and technically sound, the method is simple and practical, and importantly, the empirical evaluation is extensive and done across various benchmarks and model sizes ("The authors run extensive evaluation on models of different sizes and on different downstream tasks." - 8Ly8). Reviewers raised few questions regarding the clarity of figures and explanations, or additional studies of AGL in the zero/few-shot settings and with different PEFT methods, which appears to be sufficiently addressed during the rebuttal stage. However, there is lingering concern regarding the lack of theoretical insight or sufficiently convincing explanation on the success of AGL ("Without any explanations or theoretical insights, I am doubtful whether AGL (particularly given its origin from random initialization in a convex problem) is indeed meaningful or just a mere coincidence that has trivial explanation" - H4MK), which authors may consider addressing during paper revision stage.